# TimeMRA: LLM-Empowered Time Series Forecasting via Multi-Scale Retrieval-Augmented Representations

**Zongjiang Shang** [1 2]   **Chengxi Jin** [1 2]   **Binqing Wu** [1 2]   **Dongliang Cui** [1 2]   **Yue Yu** [1 2]   **Haobang Sun** [1 2]
**Chuanlin Xu** [1 2]   **Ling Chen** [1 2]

## Abstract

Time series forecasting plays a pivotal role in data-driven decision-making across various time series domains. Recently, leveraging their ability to extract semantically rich representations, Large Language Models (LLMs) have achieved promising results in time series forecasting. However, existing LLM-based methods struggle to obtain multi-scale retrieval-augmented representations due to entangled multi-scale representations and redundant multi-scale interference. To address this, we propose **TimeMRA**, an LLM-empowered **Time** series forecasting framework via **M**ulti-Scale **R**etrieval-**A**ugmented representations. Specifically, a scale-aware prompt generation (SAPG) module is designed to decompose time series into multiple scales and generate augmented multi-scale representations. Then, a cross-scale disentanglement constraint (CSDC) mechanism with a router network is designed to obtain the disentangled multi-scale semantic representations while mitigating interference from irrelevant scales. Finally, a cross-modality retrieval module is designed to obtain multi-scale retrieval-augmented representations for time series forecasting. Experiments on 10 real-world datasets demonstrate that TimeMRA achieves state-of-the-art (SOTA) performance.

## 1. Introduction

Time series forecasting constitutes a critical component of data-driven decision making, with wide-ranging implications across critical domains, e.g., financial forecasting (Li et al., 2024a; Qiu et al., 2025c), medical diagnostics (Wang et al., 2024; Deng et al., 2025; Fan et al., 2025), and environmental surveillance (Wu et al., 2026b;a; 2024a; Bodnar et al., 2025). The escalating complexity of modern systems has given rise to time series data characterized by non-linear temporal dependencies, high-dimensional structures, and multi-scale periodic patterns (Li et al., 2024b; Zhao et al., 2025; Chen et al., 2024a). These characteristics not only challenge traditional time series forecasting methods but also broaden the scope of contemporary deep learning-based methods, thereby enabling the adoption of emerging paradigms, e.g., Large Language Models (LLMs) (Radford et al., 2019; Touvron et al., 2023) into time series forecasting (Shang et al., 2026; Liu et al., 2025).

The progression of time series forecasting spans a broad variety of techniques. Traditional forecasting methods, e.g., ARIMA (Box & Jenkins, 1968) and Prophet (Taylor & Letham, 2018), have been proposed to capture periodic patterns, but they fail to capture non-linear relationships. To address this, deep learning-based methods, e.g., recurrent neural networks (RNNs) (Chen et al., 2021; 2025), temporal convolutional networks (TCNs) (Salinas et al., 2020; Wang et al., 2025a), graph neural networks (GNNs) (Chen et al., 2023; Cai et al., 2024), and Transformers (Cheng et al., 2024; Shang et al., 2024; Wang et al., 2025b), have been proposed for time series forecasting. Nevertheless, their performance remains constrained by the limited number of learnable parameters and small-scale training data (Jin et al., 2024; Pan et al., 2024). In contrast, benefiting from large-scale training data and adaptation techniques, recent advances (Zhou et al., 2024) provide an avenue for leveraging LLMs for time series forecasting.

Existing LLM-based methods for time series forecasting can be categorized by the function of LLMs. (1) Alignment-generative LLMs (Liu et al., 2024b; Pan et al., 2024) align input time series with the natural language modality and leverage the auto-regressive generation capability of LLMs to perform forecasting based on the aligned features. However, the inherent noise within time series and modality gap often result in modality misalignment, leading to potential semantic hallucination (Tjio et al., 2022; Shu et al., 2025).

---

[1]State Key Laboratory of Blockchain and Data Security, Zhejiang University, Hangzhou, China [2]College of Computer Science and Technology, Zhejiang University, Hangzhou, China. Correspondence to: Ling Chen <lingchen@cs.zju.edu.cn>.

*Proceedings of the $43^{rd}$ International Conference on Machine Learning*, Seoul, South Korea. PMLR 306, 2026. Copyright 2026 by the author(s).

(2) Retrieval-augmented LLMs (Cao et al., 2025; Liu et al., 2025) convert raw time series data into natural language prompts. These prompts are processed by pre-trained LLMs to yield augmented, semantically rich representations, which are then retrieved by temporal features to equip them with semantic knowledge for subsequent forecasting.

Although retrieval-augmented LLMs have achieved notable performance, they struggle to capture multi-scale representations within time series data. Specifically, while time series data naturally manifest distinct periodic patterns at different scales (Hu et al., 2025; Zhao et al., 2025), existing retrieval-augmented LLMs typically restrict learning to the original scale, thereby failing to capture complex multi-scale dynamics. However, utilizing LLMs to generate multi-scale augmented representations is non-trivial due to two primary obstacles: (1) **Entangled multi-scale representations**: To decompose input series into periodic patterns at different scales, recent studies employ series decomposition (Shang et al., 2024; Shang & Chen, 2024) and multi-periodicity analysis (Wu et al., 2022; Wang et al., 2025c) to obtain multi-scale representations. However, the multi-scale representations derived from simplistic decomposition inevitably incorporate periodic patterns from other scales, thereby undermining the independence of each scale. (2) **Redundant multi-scale interference**: Different input series prefer different scales due to their specific temporal characteristics and dynamics (Shi et al., 2024; Hu et al., 2025). Treating all multi-scale representations equally may neglect critical periodic patterns and introduce noise interference from irrelevant scales, thereby hindering LLMs in generating augmented semantic representations for time series forecasting.

Motivated by the above, we propose TimeMRA, the first LLM-empowered time series forecasting framework via multi-scale retrieval-augmented representations. Specifically, TimeMRA utilizes LLMs to generate multi-scale augmented representations, which are subsequently retrieved by temporal features through the cross-modality retrieval. The main contributions are summarized as follows:

- We design a scale-aware prompt generation (SAPG) module to decompose time series into multiple scales and generate corresponding prompts, which enables LLMs to generate augmented multi-scale representations. In addition, we introduce a router network, which can adaptively select critical multi-scale representations while mitigating interference from irrelevant scales.

- We introduce a cross-scale disentanglement constraint (CSDC) mechanism to perform cross-scale contrastive learning during the multi-modality encoder phase, which can obtain the disentangled multi-scale semantic representations for cross-modality retrieval.

- We empirically evaluate on 10 real-world datasets. The experimental results demonstrate that TimeMRA achieves state-of-the-art (SOTA) performance.

## 2. Related Work

### 2.1. Multi-Scale Modeling for Time Series

Capturing temporal dependencies across multiple scales is fundamental to time series forecasting, as real-world data often exhibits diverse periodic patterns. Multi-scale modeling for time series has evolved from classical statistical methods (Box & Jenkins, 1968; Taylor & Letham, 2018) to advanced deep learning architectures (Wen et al., 2021; Zhou et al., 2021; Wen et al., 2022; Hu et al., 2025). Early works utilized Convolutional Neural Networks (CNNs) with dilated convolutions to expand receptive fields for multi-scale feature extraction (Bai et al., 2018; Liu et al., 2022; Wu et al., 2022). Recently, Transformer-based architectures have dominated the field due to their ability to model long-range dependencies. Prominent methods such as Autoformer (Wu et al., 2021) and FEDformer (Zhou et al., 2022) integrate decomposition blocks to separately model trend and seasonal patterns. Patch-based approaches like PatchTST (Nie et al., 2022) and Pathfomer (Chen et al., 2024b) segment time series into sub-sequences to capture local semantic information while reducing computational complexity. Furthermore, Hypergraph-based methods such as MSHyper (Shang & Chen, 2024) and Ada-MSHyper (Shang et al., 2024) combine hypergraph with multi-scale structures to capture group-wise temporal pattern interactions. Despite their success, existing deep learning-based methods remain constrained by the limited number of learnable parameters and small-scale training data.

### 2.2. LLM-Based Time Series Forecasting Methods

Recent advances have demonstrated the efficacy of leveraging pre-trained LLMs for time series forecasting (Zhou et al., 2024; Pan et al., 2024). Existing LLM-based time series forecasting can be categorized into alignment-generative and retrieval-augmented methods. Alignment-generative LLMs (Jin et al., 2024; Liu et al., 2024b) leverage LLMs to perform multi-step forecasting via cross-modality alignment. Time-LLM (Liu et al., 2025) introduces a reprogramming mechanism to align the modalities between natural language and time series. AutoTimes (Liu et al., 2024b) projects time series into the language embedding space for autoregressive forecasting. However, the inherent modality gap often leads to potential semantic hallucination (Tjio et al., 2022; Shu et al., 2025). In contrast, retrieval-augmented LLMs (Liu et al., 2025; Cao et al., 2025) leverage LLMs to extract augmented semantic representations for downstream models. TEMPO (Cao et al., 2025) designs specific prompts to augment time series features. TimeCMA (Liu et al.,

2025) further combines LLM-augmented semantics with time series representations for robust forecasting. Despite their promise, these methods struggle to capture multi-scale retrieval-augmented representations within time series data.

## 3. Preliminaries

**Problem Definition.** Let $\mathbf{X} = \{\mathbf{X}_0, \ldots, \mathbf{X}_{T-1}\} \in \mathbb{R}^{T \times N}$ denote the historical time series sequence with look-back window size $T$, where $N$ represents the number of variables. At time step $t$, $\mathbf{X}_t = [x_{1,t}, \ldots, x_{N,t}]^\top$ contains the values of all variables. The goal of time series forecasting is to predict the future values sequence $\mathbf{Y} = \{\mathbf{X}_T, \ldots, \mathbf{X}_{T+H-1}\} \in \mathbb{R}^{H \times N}$ over a forecast horizon $H$.

**Fast Fourier Transform (FFT).** Given a time series sequence $\mathbf{x}_j = \{x_{j,0}, \ldots, x_{j,T-1}\} \in \mathbb{R}^T$, the FFT (Duhamel & Vetterli, 1990) maps $\mathbf{x}_j$ to a sequence of complex numbers $\mathcal{C} = \{\mathcal{C}_f\}_{f=0}^{T-1} \in \mathbb{C}^T$, where each frequency component $\mathcal{C}_f$ is formulated as follows:

$$\mathcal{C}_f = \sum_{t=0}^{T-1} x_{j,t} \cdot e^{-i\frac{2\pi}{T}ft}, \quad f = 0, \ldots, T-1 \quad (1)$$

where $f$ is the frequency index and $i$ denotes the imaginary unit ($i^2 = -1$). To quantify the intensity of each frequency component, the amplitude (or magnitude) spectrum $|C_k|$ can be obtained as follows:

$$|C_f| = \sqrt{\text{Re}(C_f)^2 + \text{Im}(C_f)^2}, \quad (2)$$

where $\text{Re}(C_f)$ and $\text{Im}(C_f)$ represent the real part and imaginary part of the magnitude spectrum, respectively. The magnitude spectrum provides a robust feature representation for identifying the dominant periodicities in time series data (Wu et al., 2022; 2025; Wang et al., 2025c).

## 4. Methodology

Figure 1 illustrates the overall framework of TimeMRA, which consists of four main components: (1) A Scale-Aware Prompt Generation (SAPG) module to initialize multi-modality inputs by integrating Multi-Scale Decomposition and Prompt Generation; (2) A Multi-Modality Encoder (MME) module to extract temporal features and disentangled multi-scale semantic representations; (3) A Cross-Modality Retrieval (CMR) module to obtain multi-scale retrieval-augmented representations; and (4) A Time Series Forecasting module to generate the final forecasting results.

### 4.1. Scale-Aware Prompt Generation (SAPG) Module

To initialize multi-modality inputs, the SAPG module first employs multi-scale decomposition to capture temporal features at different scales, and then generates the corresponding multi-scale prompts via the prompt generation.

**Multi-Scale Decomposition** Existing multi-scale decomposition methods primarily rely on frequency analysis to adaptively extract periodic patterns (Zhou et al., 2022; Wu et al., 2022; Chen et al., 2024b). However, limited by the input sequence length, they often fall short of capturing global periodic patterns and are susceptible to noise. To address this limitation, we perform frequency analysis on the entire training data to obtain the dominant periodic patterns. Specifically, given the training data $\mathbf{X}^\alpha \in \mathbb{R}^{L \times N}$, where $L$ is the length of the training data, the FFT is applied to obtain the frequency representations, which are formulated as follows:

$$\mathcal{X}^\alpha = |\mathcal{F}(\mathbf{X}^\alpha)|, \quad (3)$$

where $\mathcal{X}^\alpha \in \mathbb{R}^{L \times N}$ denotes the amplitude spectrum and $\mathcal{F}(\cdot)$ represents the FFT operation. Given that the frequency resolution of $\mathcal{X}^\alpha$ (length $L$) is higher than that of input sequence $\mathbf{X} \in \mathbb{R}^{T \times N}$ with length $T$, directly applying $\mathcal{X}^\alpha$ to $\mathbf{X}$ will lead to a frequency mismatch (Wang et al., 2025c). To resolve this, we perform spectral projection to reduce the length of $\mathcal{X}^\alpha$ from $L$ to $T$. Specifically, for each variable $\mathbf{x}_j$, the spectral projection is computed by summing $\mathcal{X}^\alpha$ over non-overlapping windows $r = L/T$, which are formulated as follows:

$$\tilde{\mathcal{X}}_{j,f}^\alpha = \sum_{k \in \Omega_f} \mathcal{X}_{j,k}^\alpha, \quad \Omega_f = [f \cdot r, (f+1) \cdot r), \quad (4)$$

where $\tilde{\mathcal{X}}_{j,f}^\alpha$ denotes the projected amplitude spectrum at the $f$-th frequency for variable $j$, with $f \in \{0, \ldots, T-1\}$. We then average the amplitude spectrum from $N$ variables to obtain a unified representation, formulated as follows:

$$\bar{\mathcal{X}}_f^\alpha = \frac{1}{N} \sum_{j=1}^{N} \tilde{\mathcal{X}}_{j,f}^\alpha. \quad (5)$$

Considering the sparsity of the frequency domain and to mitigate noise introduced by irrelevant high-frequency components, we only select the Top$S$ periodic components corresponding to the dominant spectral magnitudes, which are formulated as follows:

$$\mathcal{P} = \left\{ \frac{T}{\tau} \,\middle|\, \tau \in \text{TopS}_{f \in \{0, \ldots, \lfloor T/2 \rfloor\}} \left( \bar{X}_f^\alpha, S \right) \right\}, \quad (6)$$

where $\mathcal{P}$ denotes the set of selected dominant periodic components, and $S$ is a hyperparameter that controls the number of selected elements in the TopS operation. Due to the conjugate symmetry of the frequency domain, we only consider the frequencies within $\{0, \ldots, \lfloor T/2 \rfloor\}$. Based on the selected dominant periodic components, the subsequences $\mathbf{X}^s$ at scale $s$ are constructed as follows:

$$\mathbf{X}^s = \mathcal{G}_\phi \left( \text{Pad}(\mathbf{X}); \wp^s \right) \in \mathbb{R}^{T^s \times N}, \quad s \geq 2 \quad (7)$$

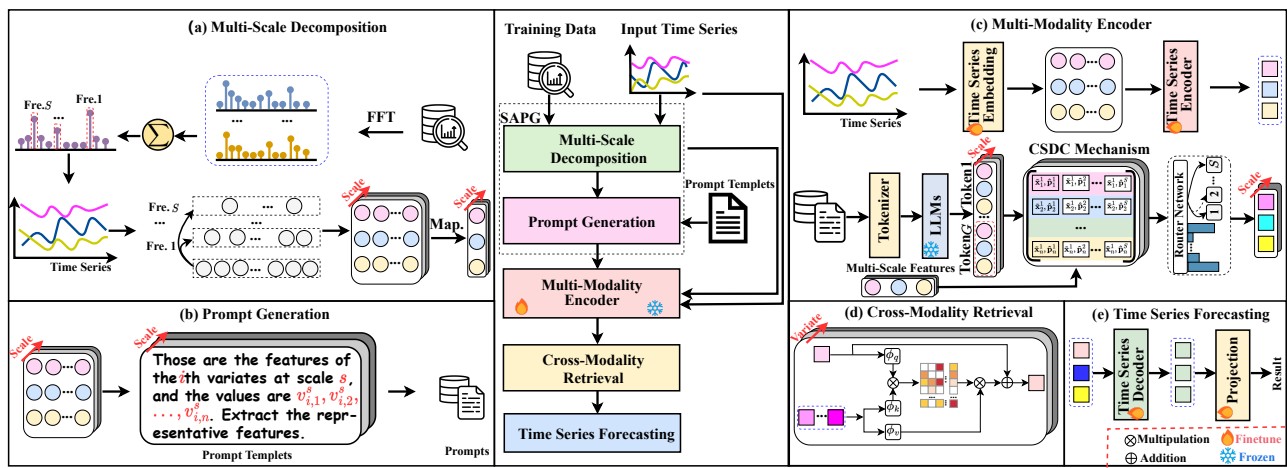

*Figure 1.* The framework of TimeMRA.

where $\mathbf{X}^1 = \mathbf{X}$ represents the input sequence. $\mathcal{G}_\phi(\cdot)$ represents a learnable aggregation layer (e.g., 1D convolution) parameterized by $\phi$. The dominant periodic component $\wp^s \in \mathcal{P}$ is used as the aggregation window size at scale $s$. $\mathrm{Pad}(\cdot)$ denotes a padding operation that ensures the aggregation produces a subsequence of length $T^s = \lceil T/\wp^s \rceil$. Then, the multi-scale temporal features at scale $s$ can be obtained by the linear mapping, which is formulated as follows:

$$\tilde{\mathbf{X}}^s = Linear(\mathbf{X}^s; \mu) \in \mathbb{R}^{C \times N}, \tag{8}$$

where $\mu$ represents the learnable parameters of the linear mapping function, and $C$ is the hidden dimension of LLMs. The final multi-scale temporal features can be represented as $\tilde{\mathbf{X}} = \{\tilde{\mathbf{X}}^1, \ldots, \tilde{\mathbf{X}}^s, \ldots, \tilde{\mathbf{X}}^S\}$.

**Prompt Generation** After obtaining subsequences at different scales, prompt templates are employed for prompt generation. As shown in Figure 1(b), the prompt $\mathbf{P}^s \in \mathbb{R}^{L^s \times N}$ at scale $s$ consists of $L^s$ elements, constructed by integrating semantic descriptions with subsequence values at scale $s$. Then, the final generated prompts can be represented as $\mathbf{P} = \{\mathbf{P}^1, \ldots, \mathbf{P}^s, \ldots, \mathbf{P}^S\}$.

### 4.2. Multi-Modality Encoder (MME) Module

The MME module is designed to capture temporal dependencies and derive augmented semantic representations via the time series encoder pathway and the LLM-augmented encoder pathway, respectively. Notably, the LLM-augmented encoder pathway uses the cross-scale disentanglement constraint (CSDC) mechanism and the router network to obtain the disentangled multi-scale representation while mitigating interference from irrelevant scales.

**Time Series Encoder Pathway** The time series encoder pathway aims to convert the input time series into variable features and capture temporal dependencies within the se-

ries. Specifically, we first employ time series embedding to transform the input time series $\mathbf{X}$. This process comprises a linear mapping followed by a single-layer normalization to enhance training stability and convergence speed, which are formulated as follows:

$$\mathbf{E} = Linear(\mathbf{X}; \alpha), \tag{9}$$
$$\tilde{\mathbf{E}} = LayerNorm(\mathbf{E}), \tag{10}$$

where $\mathbf{E} \in \mathbb{R}^{C \times N}$ indicates the variable features and $C$ is the hidden dimension. $\alpha$ denotes learnable parameters of the linear mapping function. $LayerNorm$ represents the layer normalization operation used in transformer (Liu et al., 2025) and $\tilde{\mathbf{E}} \in \mathbb{R}^{C \times N}$ is the normalized results.

Then, inspired by the Transformer architecture in LLMs, we employ a single-layer Transformer encoder as the time series encoder to capture temporal dependencies within the series, which are formulated as follows:

$$\bar{\mathbf{E}} = LayerNorm(\tilde{\mathbf{E}} + \mathrm{MHSA}(\tilde{\mathbf{E}})), \tag{11}$$
$$\hat{\mathbf{E}} = LayerNorm(\bar{\mathbf{E}} + \mathrm{FFN}(\bar{\mathbf{E}})), \tag{12}$$

where MHSA represents the multi-head self-attention mechanism and FFN refers to the feed-forward network. In addition, residual connections are integrated to combine the input and sub-layer output, followed by layer normalization to stabilize the training process.

**LLM-Augmented Encoder Pathway** As shown in Figure 1(c), the LLM-augmented encoder pathway generates augmented multi-scale representations to imbue time series with rich semantic contexts. Specifically, for the generated prompts $\mathbf{P}^s$ at scale $s$, a tokenizer is first applied to convert them into a sequence of discrete token IDs, which are then

fed into a retrieval-augmented LLM to produce augmented, semantically rich representations, formulated as follows:

$$\tilde{\mathbf{P}}^s = Tokenizer(\mathbf{P}^s), \tag{13}$$

$$\bar{\mathbf{P}}^s = LLM(\tilde{\mathbf{P}}^s), \tag{14}$$

where $\tilde{\mathbf{P}}^s \in \mathbb{R}^{G^s \times N}$ denotes the tokenlized IDs and $G^s$ is the token ID number at scale $s$. $\bar{\mathbf{P}}^s \in \mathbb{R}^{G^s \times C \times N}$ represents the augmented representations generated by LLMs. Considering the general-purpose token transformation, we froze the parameters of LLMs to reduce computational burden and align with existing literature (Lu et al., 2022; Pan et al., 2024; Zhou et al., 2024). Due to the causal masking in LLMs, the final token aggregates the most comprehensive contextual knowledge (BehnamGhader et al., 2024; Liu et al., 2025). Therefore, we retain only the final tokens $\hat{\mathbf{P}}^s \in \mathbb{R}^{C \times N}$ at each scale as the augmented representation to reduce both computational cost and information redundancy. The final augmented multi-scale semantic representations can be represented as $\hat{\mathbf{P}} = \{\hat{\mathbf{P}}^1, \ldots, \hat{\mathbf{P}}^s, \ldots, \hat{\mathbf{P}}^S\}$.

**CSDC Mechanism.** Although retrieval-augmented LLMs can transform multi-scale prompts into semantically rich representations, direct generation still faces two limitations: 1) The augmented representations generated by LLMs through rich in-context learning often lack explicit alignment with underlying numerical patterns (Pan et al., 2024; Zhao et al., 2025). 2) The entangled multi-scale representation obtained by simplistic decomposition undermines the independence of each scale. To resolve these problems, we introduce the CSDC mechanism in the LLM-augmented encoder pathway.

Specifically, given the multi-scale temporal features $\tilde{\mathbf{X}}$ and the augmented representations $\hat{\mathbf{P}}$, the proposed CSDC loss adopts a CLIP-style contrastive formulation (Radford et al., 2021) by jointly considering temporal-to-semantic and semantic-to-temporal constraints, formulated as follows:

$$\mathcal{L}_c = \frac{1}{2NS} \sum_{n=1}^{N} \sum_{s=1}^{S} \left( \ell(\tilde{\mathbf{x}}_n^s, \hat{\mathbf{p}}_n^s) + \ell(\hat{\mathbf{p}}_n^s, \tilde{\mathbf{x}}_n^s) \right), \tag{15}$$

where $\tilde{\mathbf{x}}_n^s \in \tilde{\mathbf{X}}^s$ and $\hat{\mathbf{p}}_n^s \in \hat{\mathbf{P}}^s$ represent the temporal features and semantic representations of the $n$-th variable at scale $s$, respectively. Here, $\ell(\tilde{\mathbf{x}}_n^s, \hat{\mathbf{p}}_n^s)$ is defined as a contrastive loss:

$$\ell(\tilde{\mathbf{x}}_n^s, \hat{\mathbf{p}}_n^s) = -\log \frac{\exp\left(\mathrm{sim}(\tilde{\mathbf{x}}_n^s, \hat{\mathbf{p}}_n^s)/\gamma\right)}{\sum_{k=1}^{S} \exp\left(\mathrm{sim}(\tilde{\mathbf{x}}_n^s, \hat{\mathbf{p}}_n^k)/\gamma\right)}, \tag{16}$$

where $\gamma$ is a temperature hyperparameter and $\mathrm{sim}$ represents the cosine similarity function. The reverse term $\ell(\hat{\mathbf{p}}_n^s, \tilde{\mathbf{x}}_n^s)$ is defined analogously by swapping the roles of temporal features and augmented semantic representations.

**Router Network.** After obtaining disentangled multi-scale semantic representations, a straightforward approach is to leverage them for cross-modality retrieval. However, treating all scales equally may introduce noise from less relevant scales. To address this issue, we design a Router Network that adaptively selects informative scales and suppresses redundant multi-scale interference. Specifically, for each variable $n$ at scale $s$, the scale-wise routing scores are formulated as follows:

$$g_n^s = \underset{s \in S}{Softmax}\left(\mathbf{W}[\tilde{\mathbf{x}}_n^s; \hat{\mathbf{p}}_n^s]\right), \quad \tilde{\mathbf{x}}_n^s, \hat{\mathbf{p}}_n^s \in \mathbb{R}^C \tag{17}$$

where Softmax function is performed across all scales $s \in S$, $\mathbf{W}$ are learnable parameters, and $[\cdot; \cdot]$ denotes the concatenation operation. Then, the adaptively selected mask value $\mathcal{M}_n^s \in \mathcal{M}_n$ for the $n$-th variable based on the routing scores is formulated as follows:

$$\mathcal{M}_n^s = \begin{cases} 1, & g_n^s \in \mathrm{TopM}(\{g_n^s \mid s \in S\}, M) \\ \epsilon, & \text{otherwise} \end{cases} \tag{18}$$

where $\epsilon$ is a small positive constant close to zero. $M$ denotes the quantity threshold for the TopM function. To prevent the router network from collapsing to a few scales, we introduce a load balancing loss (Fedus et al., 2022) that encourages balanced scales utilization, formulated as follows:

$$\mathcal{L}_b = \sum_{s=1}^{S} \left( \frac{1}{N} \sum_{n=1}^{N} \mathbb{I}[\mathcal{M}_n^s = 1] - \frac{M}{S} \right)^2, \tag{19}$$

where $S$ and $N$ denote the numbers of scales and variables, respectively. $\mathbb{I}[\cdot]$ is the indicator function, which equals 1 if scale $s$ is selected for variable $n$, and 0 otherwise. The loss penalizes deviations between the empirical scale usage and the uniform ratio $M/S$, encouraging balanced routing across selected scales.

### 4.3. Cross-Modality Retrieval

The cross-modality retrieval (CMR) module leverages temporal features to retrieve the augmented multi-scale semantic representations, thereby enabling the module to capture periodic patterns at different scales.

Specifically, for the given variable features $\hat{\mathbf{e}}_n \in \hat{\mathbf{E}}$ and augmented multi-scale semantic representations $\hat{\mathbf{P}}_n \in \hat{\mathbf{P}}$ of each variable, we first use three learnable matrics $\mathbf{w}_{q,n}$, $\mathbf{w}_{k,n}$, and $\mathbf{w}_{v,n}$ to transform it into query matrices $\mathbf{q}_n = \hat{\mathbf{e}}_n \mathbf{w}_{q,n}$, key matrices $\mathbf{k}_n = \hat{\mathbf{P}}_n \mathbf{w}_{k,n}$, and value matrices $\mathbf{v}_n = \hat{\mathbf{P}}_n \mathbf{w}_{v,n}$. Then, the self-attention is used to perform cross-modality retrieval, which is formulated as follows:

$$\mathbf{z}_n = Softmax\left(\frac{\mathbf{q}_n(\mathbf{k}_n)^\top}{\sqrt{d}} + log\mathcal{M}_n\right)\mathbf{v}_n, \tag{20}$$

where $\mathbf{z}_n$ represents the retrieval-augmented representations of the $n$-th variable. $\mathcal{M}_n$ represents the mask matrix of the routing network for variable $n$. Then, the final retrieval-augmented representations can be represented as $\mathcal{Z} = \{\mathbf{z}_1, \ldots, \mathbf{z}_n, \ldots, \mathbf{z}_N\}$.

### 4.4. Time Series Forecasting

The time series forecasting module aims to generate the final forecasting results and is composed of a time series decoder followed by a linear projection layer. Specifically, given the retrieval-augmented representations $\mathcal{Z}$, the time series decoder is used to capture interactions among variables, which are formulated as follows:

$$\tilde{\mathcal{Z}} = LayerNorm(\mathcal{Z} + \text{MHCA}(\mathcal{Z})), \quad (21)$$

$$\bar{\mathcal{Z}} = LayerNorm(\tilde{\mathcal{Z}} + \text{FFN}(\tilde{\mathcal{Z}})), \quad (22)$$

where $\text{MHCA}(\cdot)$ is the multi-head cross-attention that captures interactions among variables. Then, the linear projection layer is used to obtain the forecasting results, which are formulated as follows:

$$\hat{\mathbf{Y}} = Linear(\bar{Z}; \lambda) \in \mathbb{R}^{H \times N}, \quad (23)$$

where $\hat{\mathbf{Y}}$ denotes the forecasting results, and $\lambda$ is the learnable parameters of the linear projection layer.

### 4.5. Objective Function

To adaptively balance different learning objectives, we formulate the training process as a multi-task learning problem (Zhao et al., 2025). The overall loss function $\mathcal{L}$ is defined as a weighted sum over three independent loss items, which are formulated as follows:

$$\mathcal{L} = \mathcal{L}_f + \beta_1 \mathcal{L}_c + \beta_2 \mathcal{L}_b, \quad (24)$$

where $\mathcal{L}_f$ denotes the forecasting loss, and $\beta_1$ and $\beta_2$ are hyperparameters that balance different loss terms. We adopt the mean squared error (MSE) as the forecasting loss, defined as $\mathcal{L}_f = \frac{1}{T} \sum_{j=0}^{T-1} \|\mathbf{Y}_j - \hat{\mathbf{Y}}_j\|_2^2$.

### 4.6. Complexity Analysis

The time complexity of TimeMRA consists of four main parts. For the scale-aware prompt generation phase, the time complexity of the Fast Fourier Transform (FFT) is $\mathcal{O}(L \log L)$, where $L$ is the length of the training data. For the multi-modality encoder (MME) Module, the time series encoder pathway has a complexity of $\mathcal{O}(N^2)$ due to the self-attention mechanism. The LLM-augmented encoder pathway is dominated by the frozen LLM inference, with a complexity of $\mathcal{O}(S \cdot G \cdot C)$, where $G$ is the token sequence length and $C$ is the hidden dimension. For the cross-scale

*Table 1.* Dataset Statistics.

| Dataset | #Variates | Split Ratio | Frequency | Information |
|---------|-----------|-------------|-----------|-------------|
| ETTm1, ETTm2 | 7 | 6:2:2 | Hourly | Temperature |
| ETTh1, ETTh2 | 7 | 6:2:2 | 15 minutes | Temperature |
| ECL | 321 | 7:1:2 | Hourly | Electricity |
| Exchange | 8 | 7:1:2 | Daily | Economy |
| Weather | 21 | 7:1:2 | 10 minutes | Meteorology |
| PEMS-BAY | 325 | 6:2:2 | 5 minutes | Traffic Speed |
| PEMS03 | 358 | 6:2:2 | 5 minutes | Traffic Flow |
| METR-LA | 207 | 6:2:2 | 5 minutes | Traffic Speed |

disentanglement constraint (CSDC) mechanism, the complexity is $\mathcal{O}(N \cdot S^2)$. In practical operation, since FFT is precomputed only once on the training data and the number of selected scales $S$ and the routing threshold $M$ are small constants (e.g., $S = 6, M = 3$), the total complexity of TimeMRA is primarily bounded by $\mathcal{O}(N^2)$.

## 5. Experiments

**Datasets.** To evaluate the performance of our proposed method, we conduct experiments on 10 widely used real-world datasets. Specifically, for long-term time series forecasting, we perform experiments on 7 commonly used datasets, including ETT (ETTh1, ETTh2, ETTm1, and ETTm2), Weather, Electricity, Exchange-Rate, and Traffic datasets. For short-term time series forecasting, we adopt PEMS (PEMS-BAY and PEMS03) and METR-LA datasets for evaluation. In addition, for few-shot time series forecasting, we use ETT datasets for evaluation. Table 1 gives the summarized dataset statistics.

**Baselines.** We compare TimeMRA with 11 competitive baselines spanning 6 categories, including (1) Retrieval-augmented LLMs: TimeCMA (Liu et al., 2025) and UniTime (Liu et al., 2024a); (2) Alignment-generative LLMs: S$^2$IP-LLM (Pan et al., 2024) and FPT (Zhou et al., 2024); (3) Graph/Hypergraph-based methods: MSGNet (Cai et al., 2024) and MSHyper (Shang & Chen, 2024); (4) Transformer-based methods: iTransformer (Liu et al., 2023), PatchTST (Nie et al., 2022), and FEDformer (Zhou et al., 2022); (5) MLP-based methods: TimeMixer (Wang et al., 2023) and DLinear (Zeng et al., 2022); (6) CNN-based method: TimesNet (Wu et al., 2022).

**Setup.** All experiments are conducted on NVIDIA A100-80 GPUs. TimeMRA is implemented in PyTorch (Paszke et al., 2019). Adam (Kingma, 2014) is used as the optimizer. Aligning with existing methods (Liu et al., 2023; 2024a; Shang & Chen, 2024; Liu et al., 2025), we set the input length $I = 96$, with output lengths $O \in \{96, 192, 336, 720\}$ for long-term time series forecasting and few-shot forecasting, and $O \in \{12, 24, 36, 48\}$ for short-term forecasting. Mean Squared Error (MSE) and Mean Absolute Error (MAE) are set as the evaluation metrics. Neural Network

*Table 2.* Long-term forecasting results. The input length is set to 96, and the forecasting lengths are set to {96, 192, 336, 720}. Lower MSE and MAE indicate better performance. Best results are in **bold** and second best are underlined.

| Methods | | TimeMRA | | TimeCMA | | UniTime | | S²IP-LLM | | FPT | | MSHyper | | MSGNet | | TimeMxier | | iTransformer | | PatchTST | | TimesNet | | DLinear | |
|---|---|---|---|---|---|---|---|---|---|---|---|---|---|---|---|---|---|---|---|---|---|---|---|---|
| Metric | | MSE | MAE | MSE | MAE | MSE | MAE | MSE | MAE | MSE | MAE | MSE | MAE | MSE | MAE | MSE | MAE | MSE | MAE | MSE | MAE | MSE | MAE | MSE | MAE |
| ETTm1 | 96 | **0.303** | **0.347** | 0.312 | 0.351 | 0.322 | 0.363 | 0.359 | 0.381 | 0.335 | 0.369 | 0.331 | 0.350 | 0.319 | 0.366 | 0.318 | 0.356 | 0.334 | 0.368 | 0.344 | 0.373 | 0.338 | 0.375 | 0.345 | 0.372 |
| | 192 | **0.352** | **0.364** | 0.361 | 0.378 | 0.366 | 0.387 | 0.383 | 0.393 | 0.374 | 0.385 | 0.374 | 0.373 | 0.376 | 0.397 | 0.366 | 0.385 | 0.377 | 0.391 | 0.367 | 0.386 | 0.380 | 0.389 | 0.380 | 0.389 |
| | 336 | **0.365** | **0.382** | 0.392 | 0.401 | 0.398 | 0.407 | 0.416 | 0.414 | 0.407 | 0.406 | 0.408 | 0.395 | 0.417 | 0.422 | 0.396 | 0.404 | 0.426 | 0.420 | 0.392 | 0.407 | 0.410 | 0.411 | 0.413 | 0.413 |
| | 720 | **0.440** | **0.431** | 0.453 | 0.438 | 0.454 | 0.440 | 0.483 | 0.449 | 0.469 | 0.442 | 0.473 | **0.431** | 0.481 | 0.458 | 0.454 | 0.441 | 0.491 | 0.459 | 0.464 | 0.442 | 0.478 | 0.450 | 0.474 | 0.453 |
| ETTm2 | 96 | **0.163** | **0.253** | 0.173 | 0.258 | 0.183 | 0.266 | 0.193 | 0.280 | 0.190 | 0.275 | 0.179 | 0.257 | 0.177 | 0.262 | 0.175 | 0.258 | 0.180 | 0.264 | 0.177 | 0.260 | 0.187 | 0.267 | 0.193 | 0.292 |
| | 192 | **0.231** | **0.295** | 0.238 | 0.301 | 0.251 | 0.310 | 0.257 | 0.318 | 0.253 | 0.313 | 0.247 | 0.305 | 0.247 | 0.307 | 0.241 | 0.304 | 0.250 | 0.309 | 0.246 | 0.305 | 0.249 | 0.309 | 0.284 | 0.362 |
| | 336 | **0.275** | **0.331** | 0.297 | 0.338 | 0.319 | 0.351 | 0.317 | 0.353 | 0.321 | 0.360 | 0.309 | 0.344 | 0.312 | 0.346 | 0.303 | 0.343 | 0.311 | 0.348 | 0.305 | 0.343 | 0.321 | 0.351 | 0.369 | 0.427 |
| | 720 | **0.379** | **0.385** | 0.393 | 0.394 | 0.420 | 0.410 | 0.419 | 0.411 | 0.411 | 0.406 | 0.401 | 0.398 | 0.414 | 0.403 | 0.391 | 0.394 | 0.412 | 0.407 | 0.410 | 0.405 | 0.408 | 0.403 | 0.554 | 0.522 |
| ETTh1 | 96 | **0.369** | **0.375** | 0.373 | 0.391 | 0.397 | 0.418 | 0.390 | 0.400 | 0.398 | 0.424 | 0.383 | 0.392 | 0.385 | 0.411 | 0.386 | 0.405 | 0.404 | 0.413 | 0.384 | 0.402 | 0.386 | 0.400 | 0.386 | 0.400 |
| | 192 | 0.428 | 0.423 | **0.427** | **0.421** | 0.434 | 0.439 | 0.453 | 0.434 | 0.449 | 0.427 | 0.435 | 0.423 | 0.442 | 0.442 | 0.443 | 0.430 | 0.441 | 0.436 | 0.454 | 0.430 | 0.434 | 0.429 | 0.437 | 0.432 |
| | 336 | **0.426** | **0.431** | 0.458 | 0.448 | 0.468 | 0.457 | 0.461 | 0.449 | 0.492 | 0.466 | 0.480 | 0.445 | 0.480 | 0.468 | 0.512 | 0.470 | 0.487 | 0.458 | 0.497 | 0.462 | 0.491 | 0.469 | 0.481 | 0.459 |
| | 720 | **0.443** | **0.447** | 0.449 | 0.460 | 0.469 | 0.477 | 0.469 | 0.470 | 0.487 | 0.483 | 0.482 | 0.467 | 0.494 | 0.488 | 0.498 | 0.476 | 0.496 | 0.481 | 0.496 | 0.481 | 0.521 | 0.500 | 0.519 | 0.516 |
| ETTh2 | 96 | **0.275** | **0.325** | 0.286 | 0.336 | 0.296 | 0.345 | 0.295 | 0.345 | 0.312 | 0.360 | 0.291 | 0.338 | 0.328 | 0.371 | 0.296 | 0.347 | 0.297 | 0.349 | 0.312 | 0.358 | 0.340 | 0.374 | 0.333 | 0.387 |
| | 192 | **0.355** | **0.371** | 0.363 | 0.387 | 0.374 | 0.394 | 0.386 | 0.399 | 0.387 | 0.405 | 0.376 | 0.391 | 0.402 | 0.414 | 0.376 | 0.394 | 0.380 | 0.400 | 0.397 | 0.408 | 0.402 | 0.414 | 0.477 | 0.476 |
| | 336 | 0.414 | 0.427 | **0.406** | **0.421** | 0.415 | 0.427 | 0.419 | 0.429 | 0.424 | 0.437 | 0.414 | 0.430 | 0.424 | 0.443 | 0.434 | 0.443 | 0.428 | 0.432 | 0.435 | 0.440 | 0.452 | 0.452 | 0.594 | 0.541 |
| | 720 | **0.417** | **0.435** | **0.417** | 0.438 | 0.425 | 0.444 | 0.425 | 0.442 | 0.433 | 0.453 | **0.417** | **0.435** | 0.417 | 0.441 | 0.464 | 0.464 | 0.427 | 0.445 | 0.436 | 0.449 | 0.462 | 0.468 | 0.831 | 0.657 |
| ECL | 96 | **0.129** | **0.233** | 0.143 | 0.238 | 0.196 | 0.287 | 0.172 | 0.265 | 0.197 | 0.290 | 0.152 | 0.252 | 0.165 | 0.274 | 0.153 | 0.247 | 0.148 | 0.240 | 0.186 | 0.269 | 0.168 | 0.272 | 0.197 | 0.282 |
| | 192 | **0.155** | **0.241** | 0.161 | 0.259 | 0.199 | 0.291 | 0.182 | 0.279 | 0.201 | 0.292 | 0.171 | 0.271 | 0.184 | 0.292 | 0.166 | 0.256 | 0.162 | 0.253 | 0.190 | 0.273 | 0.184 | 0.289 | 0.196 | 0.285 |
| | 336 | **0.157** | 0.263 | 0.169 | **0.261** | 0.214 | 0.305 | 0.195 | 0.288 | 0.217 | 0.309 | 0.187 | 0.284 | 0.195 | 0.303 | 0.185 | 0.277 | 0.178 | 0.269 | 0.206 | 0.290 | 0.198 | 0.300 | 0.209 | 0.301 |
| | 720 | 0.224 | **0.298** | **0.219** | 0.315 | 0.254 | 0.335 | 0.233 | 0.320 | 0.253 | 0.339 | 0.224 | 0.316 | 0.231 | 0.332 | 0.225 | 0.310 | 0.225 | 0.317 | 0.247 | 0.322 | 0.220 | 0.320 | 0.245 | 0.333 |
| Exchange | 96 | **0.081** | **0.198** | 0.093 | 0.214 | 0.086 | 0.209 | 0.083 | 0.201 | 0.091 | 0.212 | 0.083 | 0.199 | 0.102 | 0.230 | 0.093 | 0.214 | 0.086 | 0.206 | 0.088 | 0.205 | 0.107 | 0.234 | 0.088 | 0.218 |
| | 192 | 0.172 | 0.295 | 0.209 | 0.329 | 0.174 | 0.299 | **0.170** | **0.293** | 0.183 | 0.304 | 0.173 | 0.294 | 0.195 | 0.317 | 0.215 | 0.326 | 0.177 | 0.299 | 0.176 | 0.299 | 0.226 | 0.344 | 0.176 | 0.315 |
| | 336 | 0.305 | **0.389** | 0.327 | 0.425 | 0.319 | 0.408 | 0.333 | 0.417 | 0.328 | 0.417 | 0.310 | 0.411 | 0.359 | 0.436 | 0.385 | 0.451 | 0.331 | 0.417 | **0.301** | 0.397 | 0.367 | 0.448 | 0.313 | 0.427 |
| | 720 | **0.831** | **0.681** | 0.877 | 0.705 | 0.875 | 0.701 | 0.875 | 0.706 | 0.880 | 0.704 | 0.846 | 0.692 | 0.940 | 0.738 | 0.915 | 0.957 | 0.847 | 0.691 | 0.901 | 0.714 | 0.964 | 0.746 | 0.839 | 0.695 |
| Weather | 96 | 0.159 | **0.187** | 0.167 | 0.211 | 0.171 | 0.214 | 0.198 | 0.235 | 0.203 | 0.244 | **0.157** | 0.198 | 0.165 | 0.212 | 0.163 | 0.210 | 0.174 | 0.214 | 0.177 | 0.218 | 0.172 | 0.220 | 0.196 | 0.255 |
| | 192 | 0.216 | 0.261 | 0.212 | 0.253 | 0.217 | 0.254 | 0.240 | 0.269 | 0.247 | 0.277 | **0.207** | **0.244** | 0.212 | 0.254 | 0.212 | 0.257 | 0.221 | 0.254 | 0.222 | 0.259 | 0.219 | 0.261 | 0.237 | 0.296 |
| | 336 | **0.247** | **0.276** | 0.270 | 0.292 | 0.274 | 0.293 | 0.295 | 0.308 | 0.297 | 0.311 | 0.265 | 0.286 | 0.272 | 0.299 | 0.263 | 0.292 | 0.278 | 0.296 | 0.277 | 0.297 | 0.280 | 0.306 | 0.283 | 0.335 |
| | 720 | **0.334** | **0.317** | 0.350 | 0.348 | 0.351 | 0.343 | 0.368 | 0.353 | 0.368 | 0.356 | 0.342 | 0.339 | 0.350 | 0.348 | 0.343 | 0.345 | 0.358 | 0.349 | 0.352 | 0.347 | 0.365 | 0.359 | 0.345 | 0.381 |

*Table 3.* Short-term forecasting results. The input length is set to 96, and the forecasting results are averaged from {12, 24, 36, 48}. Lower values indicate better performance. Best results are in **bold** and second best are underlined. Full results are given in Appendix E, Table 10.

| Methods | TimeMRA | | TimeCMA | | UniTime | | S²IP-LLM | | FPT | | MSHyper | | MSGNet | | TimeMixer | | iTransformer | | PatchTST | | TimesNet | | DLinear | |
|---|---|---|---|---|---|---|---|---|---|---|---|---|---|---|---|---|---|---|---|---|---|---|---|---|
| Metric | MSE | MAE | MSE | MAE | MSE | MAE | MSE | MAE | MSE | MAE | MSE | MAE | MSE | MAE | MSE | MAE | MSE | MAE | MSE | MAE | MSE | MAE | MSE | MAE |
| PEMS03 | **0.138** | **0.239** | 0.160 | 0.262 | 0.226 | 0.351 | 0.199 | 0.323 | 0.231 | 0.315 | 0.305 | 0.351 | 1.269 | 0.941 | 0.308 | 0.367 | 0.370 | 0.363 | 0.279 | 0.358 | 1.233 | 0.893 | 0.217 | 0.323 |
| PEMS-BAY | **0.658** | **0.369** | 0.683 | 0.394 | 0.701 | 0.403 | 0.710 | 0.405 | 0.782 | 0.427 | 0.787 | 0.405 | 1.284 | 0.937 | 0.771 | 0.415 | 0.923 | 0.481 | 0.774 | 0.417 | 1.417 | 0.651 | 0.726 | 0.428 |
| METR-LA | **0.623** | **0.389** | 0.638 | 0.399 | 0.657 | 0.409 | 0.675 | 0.445 | 0.715 | 0.445 | 0.727 | 0.409 | 1.123 | 0.658 | 0.692 | 0.442 | 1.086 | 0.637 | 0.691 | 0.449 | 0.980 | 0.584 | 0.648 | 0.467 |

Intelligence (NNI)[1] is used to automatically optimize the hyperparameters, and the source code is made available on GitHub[2]. More details about datasets, baselines, and experimental settings are given in Appendix A, B, and C, respectively.

## 5.1. Main Results

**Long-Term Time Series Forecasting**. Table 2 presents the long-range time series forecasting results. We can observe that: (1) LLM4TS methods (i.e., TimeCMA, UniTime, S²IP-LLM, and FPT) perform better than other deep learning-based and linear-based methods. This confirms that the extensive pre-trained knowledge and sophisticated pattern extraction capabilities of LLMs can be transferred to the time series modality. (2) Among LLM4TS methods, benefiting from semantically rich representation extraction ability, retrieval-augmented LLMs demonstrate superior performance compared to alignment-generative LLMs. However, they are challenged by the multi-scale representation within time series data. (3) By considering multi-scale augmented representations, TimeMRA consistently outperforms existing LLM4TS methods across nearly all settings. Specifically, TimeMRA reduces forecasting error by an average of 4.04% and 3.69% compared to the second best baselines in

terms of MSE and MAE, respectively.

**Short-Term Time Series Forecasting**. Table 3 summarizes the results of short-term time series forecasting. It is notable that METR-LA and PEMS (i.e., PEMS-BAY and PEMS03) datasets consist of multiple traffic flow series collected from citywide transportation networks, exhibiting complex spatial–temporal dependencies across different variables. We adopt the same settings as existing methods (Liu et al., 2023; Wang et al., 2023). TimeMRA still delivers superior performance compared to existing methods, possibly because the cross-modality retrieval model captures complex multi-scale semantic representations, while the time series forecasting model exploits dependencies among variables. Specifically, TimeMRA achieves performance improvements of up to 6.64% in MSE and 5.85% in MAE compared to the second best baselines.

**Few-Shot Time Series Forecasting**. LLM4TS methods excel in diverse forecasting tasks, particularly in scenarios with limited training data. We use 5% and 10% training data on ETT datasets to evaluate the few-shot learning performance. Table 4 shows the few-shot forecasting results. We observe that retrieval-augmented LLMs perform better than alignment-generative LLMs. The reason may be that under limited training data scenarios, the inherent noise within time series may lead to potential semantic hallucination, which interferes with time series forecasting. In contrast, retrieval-augmented LLMs can extract augmented semantic

---

[1] https://nni.readthedocs.io/en/latest/
[2] https://github.com/shangzongjiang/TimeMRA/

*Table 4.* Few-shot forecasting results. All results are averaged from four different forecasting lengths {96, 192, 336, 720}. Lower values indicate better performance. Best results are in **bold** and second best are underlined. Full results are given in Table 11 and Table 12.

| Dataset | 5% training data | | | | | | | | 10% training data | | | | | | | |
|---|---|---|---|---|---|---|---|---|---|---|---|---|---|---|---|---|
| | ETTh1 | | ETTh2 | | ETTm1 | | ETTm2 | | ETTh1 | | ETTh2 | | ETTm1 | | ETTm2 | |
| Metric | MSE | MAE | MSE | MAE | MSE | MAE | MSE | MAE | MSE | MAE | MSE | MAE | MSE | MAE | MSE | MAE |
| FPT | 0.669 | 0.556 | 0.528 | 0.477 | 0.645 | 0.525 | 0.317 | 0.350 | 0.604 | 0.526 | 0.587 | 0.501 | 0.604 | 0.499 | 0.303 | 0.337 |
| S$^2$IP-LLM | 0.943 | 0.675 | 0.522 | 0.484 | 0.720 | 0.572 | 0.311 | 0.354 | 0.745 | 0.584 | 0.504 | 0.474 | 0.578 | 0.494 | 0.303 | 0.338 |
| UniTime | 0.719 | 0.560 | 0.417 | 0.423 | 0.611 | 0.496 | 0.303 | 0.337 | 0.695 | 0.562 | 0.419 | 0.429 | 0.531 | 0.467 | 0.299 | 0.335 |
| TimeCMA | 0.679 | 0.582 | 0.416 | 0.414 | 0.648 | 0.527 | 0.324 | 0.360 | 0.589 | 0.528 | 0.459 | 0.454 | 0.609 | 0.511 | 0.309 | 0.344 |
| TimeMRA | **0.629** | **0.507** | **0.390** | **0.399** | **0.591** | **0.477** | **0.283** | **0.328** | **0.565** | **0.496** | **0.390** | **0.406** | **0.515** | **0.449** | **0.277** | **0.324** |

*Table 5.* Results of different SAPG modules and CSDC mechanism on ETTh1 dataset. The best results are **bolded**.

| Methods | w/o SAPG | -PAW | w/o CSDC | w/o TSC | w/o STC | TimeMRA |
|---|---|---|---|---|---|---|
| Metric | MSE MAE | MSE MAE | MSE MAE | MSE MAE | MSE MAE | MSE MAE |
| 96 | 0.372 0.385 | 0.373 0.381 | 0.422 0.429 | 0.417 0.394 | 0.407 0.382 | **0.369 0.375** |
| 192 | 0.435 0.439 | 0.433 0.427 | 0.456 0.453 | 0.457 0.442 | 0.437 0.443 | **0.428 0.423** |
| 336 | 0.441 0.447 | 0.434 0.437 | 0.490 0.466 | 0.462 0.449 | 0.435 0.451 | **0.426 0.431** |

*Table 6.* Results of different LLM backbones on ETTh1 dataset. The best results are **bolded**.

| Methods | GPT-M | L-7B | Q-7B | D-8B | TimeMRA |
|---|---|---|---|---|---|
| Metric | MSE MAE | MSE MAE | MSE MAE | MSE MAE | MSE MAE |
| 96 | 0.365 0.343 | **0.353** 0.337 | 0.358 0.348 | 0.354 **0.331** | 0.369 0.347 |
| 192 | 0.423 0.416 | 0.406 **0.409** | 0.409 0.411 | **0.403** 0.413 | 0.428 0.423 |
| 336 | 0.433 0.427 | 0.429 0.430 | 0.418 **0.423** | **0.411** 0.426 | 0.426 0.431 |

representations from limited datasets. Notably, TimeMRA achieves the best performance in all datasets, giving an average of 10.05% and 8.72% error reductions compared to the second best baselines in terms of MSE and MAE, respectively. This is because by leveraging the multi-modality encoder (MME) module and cross-modality retrieval module, TimeMRA can obtain multi-scale retrieval-augmented representations that are beneficial for forecasting.

### 5.2. Ablation Studies

**SAPG Module**. To assess the effectiveness of the SAPG module, we conduct ablation studies by designing the following variants: (1) Removing the SAPG module and directly feeding the raw input sequence into the multi-modality encoder (w/o SAPG). (2) Replacing the multi-scale decomposition module within SAPG module with predefined multi-scale aggregation windows (-PAW). The experimental results are shown in Table 5, where w/o SAPG exhibits the worst performance, showing the effectiveness of the SAPG module. In addition, -PAW performs worse than TimeMRA, likely because predefined aggregation windows cannot adaptively capture multi-scale periodic patterns.

**CSDC mechanism**. To investigate the effectiveness of CSDC mechanism used in the multi-modality encoder module, we conduct the following variants: (1) Removing the CSDC mechanism (w/o CSDC). (2) Removing the temporal-to-semantic constraints used in the CSDC mechanism (w/o TSC). (3) Removing the semantic-to-temporal constraints used in the CSDC mechanism (w/o STC). The experimental results on ETTh1 dataset are shown in Table 5. We can observe that TimeMRA performs better than R.4 and R.5, demonstrating the effectiveness of the temporal-to-semantic and semantic-to-temporal constraints, respectively. In addition, w/o CSDC yields the worst performance, which may be attributed to the fact that without the CSDC mechanism, the entangled multi-scale representation undermines the in-

dependence of each scale and lacks explicit alignment with numerical patterns.

**LLM backbones.** For a fair comparison, we adopt GPT2-small as the default LLM backbone. However, TimeMRA can adopt more advanced LLMs for augmented multi-scale representations generation. To investigate the impact of different LLM backbones, we adopt GPT2-medium, LLaMA2-7B, Qwen2.5-7B, and DeepSeekR1-8B for comparison. The experimental results on ETTh1 dataset with input length $I = 96$ and output length $O = 96$ are shown in Table 6. TimeMRA achieves better performance with more advanced LLM backbones. The reason is that more advanced LLMs possess superior semantic understanding capabilities, which enable the model to produce more informative and expressive multi-scale augmented representations.

**Multi-Modality Encoder(MME) Module**. To investigate the effectiveness of the MME module, we conduct ablation studies by designing the following variants: (1) Replacing the time series encoder pathway with simple linear layers (-R TSE); (2) Replacing the LLM used in the LLM-augmented encoder pathway with simple linear layers (-R LLM). The experimental results on ETTh1 dataset are shown in Table 7, from which we can observe that TimeMRA performs better than -R TSE, showing the effectiveness of the time series encoder pathway. In addition, we observe that the -R LLM exhibits the worst performance. This may be due to the absence of LLM, which prevents the module from extracting semantically rich representations. The experimental results highlight the effectiveness of LLM in extracting semantically rich representations.

**Cross-Modality Retrieval (CMR) Module**. To investigate the effectiveness of the CMR module, we conduct ablation studies by designing the following variant: Removing the CMR module, directly concatenating the variable features with the augmented multi-scale semantic representations, and feeding them into a linear layer to obtain the retrieval-

*Table 7.* Results of different MME, CMR, and time series forecasting modules on ETTh1 dataset. The best results are **bolded**.

| Methods | -R TSE | | -R LLM | | -w/o CMR | | -R TSF | | TimeMRA | |
|---------|--------|--------|--------|--------|----------|--------|--------|--------|---------|--------|
| Metric | MSE | MAE | MSE | MAE | MSE | MAE | MSE | MAE | MSE | MAE |
| 96 | 0.377 | 0.380 | 0.400 | 0.393 | 0.399 | 0.383 | 0.374 | 0.382 | **0.369** | **0.375** |
| 192 | 0.429 | 0.433 | 0.448 | 0.457 | 0.437 | 0.443 | 0.437 | 0.441 | **0.428** | **0.423** |
| 336 | 0.433 | 0.437 | 0.486 | 0.466 | 0.445 | 0.434 | 0.441 | 0.445 | **0.426** | **0.431** |

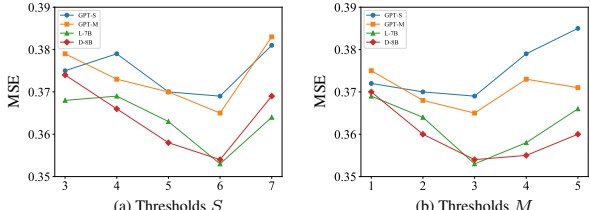

*Figure 2.* The impact of different hyperparameters.

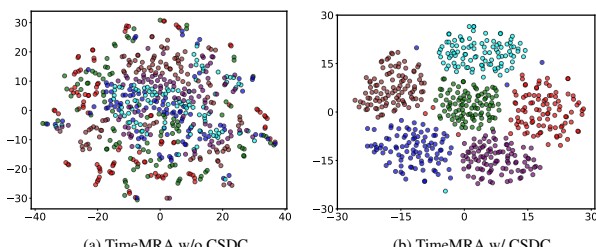

*Figure 3.* t-SNE results of multi-scale semantic representations.

augmented representations (-w/o CMR). The experimental results on ETTh1 dataset are shown in Table 7. We observe that -w/o CMR performs worse than TimeMRA, showing the effectiveness of the CMR module in capturing retrieval-augmented representations.

**Time Series Forecasting Module**. To investigate the effectiveness of the time series forecasting module, we conduct ablation studies by designing the following variant: replacing the time series forecasting module with simple linear layers (-R TSF). The experimental results on ETTh1 dataset are shown in Table 7. We observe that TimeMRA performs better than -R TSF, showing the effectiveness of the time series forecasting module. This can be attributed to the fact that, compared to the time series forecasting module, a simple linear mapping is unable to capture the complex spatial interactions among variables.

### 5.3. Parameter Studies

TimeMRA leverages TopS and TopM functions to determine the dominant spectral magnitude and select the number of scales in the Router Network. We perform parameter studies to investigate the impact of the thresholds $S$ and $M$. As illustrated in Figure 2, experiments are performed on ETTh1 dataset with an output length $O = 96$, using GPT2-small, GPT2-medium, LLaMA2-7B, and DeepSeekR1-8B as LLM backbones. We observe that the best performance is achieved when $S = 6$. The reason may be that a smaller $S$ discards useful information, while a larger $S$ dilutes the contribution of dominant frequencies and introduces noise interference. In addition, $M = 3$ yields the best performance. This is because a small $M$ restricts the routing network to only a few scales, preventing it from fully exploiting diverse periodic patterns, while an excessively large $M$ weakens the selectivity of the routing network. Notably, even using different LLM backbones, the optimal values for the dominant spectral magnitude and the number of scales remain consistent. This suggests that these choices are primarily driven by the inherent periodic patterns of the time series, which are independent of the specific LLM employed.

### 5.4. Qualitative Analysis

The ablation studies in Table 5 indicate that removing the CSDC mechanism causes a substantial drop in performance. To further investigate its impact, we conduct a qualita-

tive analysis on ETTh1 dataset using t-SNE visualization. Specifically, Figure 3 illustrates the multi-scale semantic representations of 100 randomly selected samples generated with and without the CSDC mechanism. From Figure 3(a), we can observe that without the CSDC mechanism, the multi-scale semantic representations are entangled and lack clear clustering. In contrast, Figure 3(b) illustrates that, with the integration of the CSDC mechanism, the semantic representations at different scales are grouped into distinct clusters. This highlights the effectiveness of CSDC in producing disentangled multi-scale semantic representations.

## 6. Conclusions

This paper proposed TimeMRA, the first LLM-empowered time series forecasting framework via multi-scale retrieval-augmented representations. Specifically, a scale-aware prompt generation (SAPG) module is designed to generate augmented multi-scale representations and a cross-scale disentanglement constraint (CSDC) mechanism is used to obtain the disentangled multi-scale semantic representations for cross-modality retrieval. Experiment results and qualitative analysis justify the effectiveness of TimeMRA.

## Impact Statement

This study focuses on time series forecasting, aiming to address the issues of entangled multi-scale representations and redundant multi-scale interference. In addition, this paper aims to advance the field of Machine Learning. Since our work focuses solely on scientific issues and uses only publicly available datasets, we believe there are no ethical risks associated with our research.

## Acknowledgment

This work was supported by the "Pioneer" R&D Program of Zhejiang (Grant No. 2026C01015).

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

# A. Dataset Details

We extensively evaluate the proposed TimeMRA on 10 commonly used real-world datasets, including ETT (ETTh1, ETTh2, ETTm1, and ETTm2), Weather, Electricity, Exchange-Rate, Traffic, PEMS (PEMS-BAY and PEMS03), and METR-LA datasets. The details about the 10 real-world datasets are given as follows:

**ETT** (Zhou et al., 2021): The Electricity Transformer Temperature (ETT) dataset is collected from two distinct counties in China, consisting of ETTh (ETTh1 and ETTh2) and ETTm (ETTm1 and ETTm2) datasets, with sampling frequencies of 1 hour and 15 minutes, respectively. Each dataset includes six power load features and one target "oil temperature" feature.

**Weather**[3]: The Weather dataset is collected from the Weather Station at the Max Planck Institute for Biogeochemistry. It includes 21 meteorological variables (e.g., air temperature and humidity), with a sampling frequency of 10 minutes.

**Exchange-Rate** (Lai et al., 2018): The Exchange-Rate (Exchange) dataset consists of historical exchange rates from 8 countries, with a sampling frequency of 1 day.

**Electricity**[4]: The Electricity (ECL) dataset contains the hourly electricity consumption of 321 customers from the UCI Machine Learning Repository. The sampling frequency is 1 hour.

**PEMS** (Chen et al., 2023; 2001): The PEMS dataset is collected by the California Transportation Agencies, with the PEMS-BAY dataset containing traffic speed data measured by 325 sensors and the PEMS03 dataset containing traffic flow measured by 358 sensors, respectively. The sampling frequency is 5 minutes.

**METR-LA** (Chen et al., 2024a): The METR-LA dataset contains the average traffic speeds from 207 sensors across Los Angeles, with a sampling frequency of 5 minutes.

Following existing works (Shang & Chen, 2024; Liu et al., 2025; Qiu et al., 2024; 2025b;a), each dataset is split into training, validation, and testing sets based on the chronological order. For ETT (ETTh1, ETTh2, ETTm1, and ETTm2), PEMS (PEMS-BAY and PEMS03), and METR-LA dataset, the train-validation-test split ratio is 6:2:2. For ECL, Exchange, and Weather dataset, the train-validation-test split ratio is 7:1:2.

# B. Methods for Comparison

We compared TimeMRA with 11 competitive baselines spanning 6 categories. The baselines used for comparison are given as follows:

**Retrieval-Augmented LLMs**:

TimeCMA (Liu et al., 2025): It presents a dual-modality encoding with two branches: A time series encoding branch that extracts time series embeddings, and an LLM-empowered encoding branch that produces retrieval-augmented representations. Cross-modality alignment and retrieval are then applied to obtain semantically rich time-series embeddings for forecasting.

UniTime(Liu et al., 2024a): It trains a model across multiple time series domains, using domain instructions and a language-TS Transformer to offer retrieval-augmented representations for cross-domain time series forecasting.

**Alignment-Generative LLMs**:

$S^2$IP-LLM (Pan et al., 2024): It decomposes the input time series into multiple components and constructs time-series embeddings using patching operations. Then it leverages word token embeddings from pre-trained LLMs to derive semantic anchors and aligns selected anchors with the time-series embeddings by maximizing cosine similarity in a joint space.

FPT (Zhou et al., 2024): It transforms a frozen pre-trained language model to form a unified time-series model that generalizes across datasets, tasks, and forecasting horizons without task-specific customization.

**Graph/Hypergraph-Based Methods**:

MSGNet (Cai et al., 2024): It performs frequency analysis on the input sequence to extract dominant periodic patterns and performs multi-scale decomposition according to the dominant periodic patterns. Then, it combines GNNs with the self-attention mechanism to inter-series correlation and intra-series dependencies.

---

[3]https://www.bgc-jena.mpg.de/wetter/
[4]https://archive.ics.uci.edu/ml/datasets/ElectricityLoadDiagrams20112014

*Table 8.* Settings of NNI.

|  | Parameters | Choise |
|---|---|---|
| Search space | Batch size | {8, 16, 32, 64, 128} |
|  | Dropout rate | {0.05, 0.1, 0.5} |
|  | Threshold $S$ | {3, 4, 5, 6, 7} |
|  | Threshold $M$ | {1, 2, 3, 4, 5} |
| Configures | Max trial number | 120 |
|  | Optimization algorithm | Tree-structured Parzen Estimator |
|  | Early stopping strategy | Curvefitting |

MSHyper (Shang & Chen, 2024): It designs a multi-scale hypergraph to provide foundations for modeling group-wise pattern interactions and introduces a hyperedge graph to enhance hypergraph modeling. In addition, a tri-stage message passing mechanism is used to learn the interaction strength between periodic patterns of difference scales.

**Transformer-Based Methods**:

iTransformer (Liu et al., 2023): It converts the time points of individual series into different variate tokens, and subsequently applies attention mechanisms and a feed-forward network to model multivariate correlations and nonlinear temporal dependencies, respectively.

PathchTST (Nie et al., 2022): It segments time series into subsequence-level patches and treats them as input tokens for Transformers to model temporal dependencies in a channel-independent manner.

FEDformer (Zhou et al., 2022): It introduces a seasonal-trend decomposition method to capture the global profile of time series and uses a Transformer framework to capture more detailed structures.

**MLP-Based Methods**:

TimeMixer (Wang et al., 2023): It defines multiple scales in the time domain and various resolutions in the frequency domain. Then, it employs various mixing strategies to extract intricate, task-adaptive time series patterns.

DLinear (Zeng et al., 2022): It decomposes input sequences into periodic patterns at different scales and employs a series of simple one-layer models to capture the temporal dependencies of different components.

**CNN-Based Methods**:

TimesNet (Wu et al., 2022): It performs frequency analysis on the input time series to identify multiple periods, and transforms the 1D time series into 2D tensors according to the dominant periods. It then applies 2D convolution kernels to capture both intraperiod- and interperiod-variations.

## C. Experimental Settings

TimeMRA is implemented in PyTorch (Paszke et al., 2019). For a fair comparison, we use GPT2-Small (Radford et al., 2019) as the default LLM backbone unless specified otherwise. We repeat all experiments 3 times and use the mean as the final result. Adam (Kingma, 2014) is used as the optimizer with the initial learning rate range from $10^{-3}$ to $10^{-1}$. MSE is used as the loss function. Following existing works (Liu et al., 2024a; Shang & Chen, 2024; Liu et al., 2025), we set the input length $I = 96$, with output length $O \in \{96, 192, 336, 720\}$ for long-term time series forecasting and few-shot forecasting, and $O \in \{12, 24, 36, 48\}$ for short-term forecasting. For other hyperparameters, we utilize the Neural Network Intelligence (NNI)[5] toolkit to automatically search for the optimal values. Compared to grid search, the NNI toolkit significantly reduces computational costs. The detailed search space for the key hyperparameters is provided in Table 8. The source code for TimeMRA is available on GitHub[6].

## D. Evaluation Metrics

Following existing methods (Liu et al., 2023; 2025; Zhang et al., 2025; Wu et al., 2024b), we adopt Mean Squared Error (MSE) and Mean Absolute Error (MAE) as evaluation metrics, which are defined as follows:

---

[5] https://nni.readthedocs.io/en/latest/
[6] https://github.com/shangzongjiang/TimeMRA/

*Table 9.* Long-term time series forecasting results. The input length is set to 96, and the forecasting results are averaged from {96, 192, 336, 720}. Lower MSE and MAE indicate better performance. Best results are in **bold** and the second best are underlined.

| Methods | TimeMRA | | TimeCMA | | S²IP-LLM* | | UniTime | | FPT | | MSHyper* | | MSGNet* | | TimeMxier | | iTransformer | | PatchTST | | TimesNet | | DLinear | |
|---|---|---|---|---|---|---|---|---|---|---|---|---|---|---|---|---|---|---|---|---|---|---|---|---|
| Metric | MSE | MAE | MSE | MAE | MSE | MAE | MSE | MAE | MSE | MAE | MSE | MAE | MSE | MAE | MSE | MAE | MSE | MAE | MSE | MAE | MSE | MAE | MSE | MAE |
| ETTm1 | **0.365** | **0.381** | 0.380 | 0.392 | 0.410 | 0.409 | 0.385 | 0.399 | 0.396 | 0.401 | 0.397 | 0.387 | 0.398 | 0.411 | 0.384 | 0.397 | 0.407 | 0.410 | 0.392 | 0.402 | 0.400 | 0.406 | 0.403 | 0.407 |
| ETTm2 | **0.262** | **0.316** | 0.275 | 0.323 | 0.296 | 0.340 | 0.293 | 0.334 | 0.294 | 0.339 | 0.284 | 0.326 | 0.288 | 0.330 | 0.278 | 0.325 | 0.288 | 0.332 | 0.285 | 0.328 | 0.291 | 0.333 | 0.350 | 0.401 |
| ETTh1 | **0.418** | **0.419** | 0.423 | 0.431 | 0.448 | 0.443 | 0.442 | 0.448 | 0.457 | 0.450 | 0.445 | 0.432 | 0.452 | 0.452 | 0.460 | 0.445 | 0.454 | 0.447 | 0.463 | 0.449 | 0.458 | 0.450 | 0.456 | 0.452 |
| ETTh2 | **0.365** | **0.390** | 0.372 | 0.397 | 0.381 | 0.404 | 0.378 | 0.403 | 0.389 | 0.414 | 0.376 | 0.399 | 0.396 | 0.417 | 0.393 | 0.412 | 0.383 | 0.407 | 0.395 | 0.414 | 0.414 | 0.427 | 0.559 | 0.515 |
| ECL | **0.166** | **0.259** | 0.174 | 0.269 | 0.195 | 0.288 | 0.216 | 0.306 | 0.217 | 0.308 | 0.184 | 0.281 | 0.194 | 0.300 | 0.182 | 0.273 | 0.178 | 0.270 | 0.207 | 0.289 | 0.192 | 0.295 | 0.212 | 0.300 |
| Exchange* | **0.347** | **0.391** | 0.377 | 0.418 | 0.365 | 0.404 | 0.364 | 0.404 | 0.371 | 0.409 | 0.353 | 0.399 | 0.399 | 0.430 | 0.402 | 0.486 | 0.360 | 0.403 | 0.367 | 0.404 | 0.416 | 0.443 | 0.354 | 0.414 |
| Weather | **0.239** | **0.260** | 0.250 | 0.276 | 0.275 | 0.291 | 0.253 | 0.276 | 0.279 | 0.297 | 0.243 | 0.267 | 0.249 | 0.278 | 0.245 | 0.276 | 0.258 | 0.278 | 0.257 | 0.280 | 0.259 | 0.287 | 0.265 | 0.317 |

*Table 10.* Full results of short-term time series forecasting. The input length is set to 96, and the forecasting lengths are set to {12, 24, 36, 48}. Lower MSE and MAE indicate better performance. Best results are in **bold** and the second best are underlined.

| Methods | | TimeMRA | | TimeCMA | | UniTime | | S²IP-LLM | | FPT | | MSHyper | | MSGNet | | TimeMixer | | iTransformer | | PatchTST | | TimesNet | | DLinear | |
|---|---|---|---|---|---|---|---|---|---|---|---|---|---|---|---|---|---|---|---|---|---|---|---|---|---|
| Metrics | | MSE | MAE | MSE | MAE | MSE | MAE | MSE | MAE | MSE | MAE | MSE | MAE | MSE | MAE | MSE | MAE | MSE | MAE | MSE | MAE | MSE | MAE | MSE | MAE |
| PEMS03 | 12 | **0.054** | **0.147** | 0.065 | 0.179 | 0.132 | 0.275 | 0.094 | 0.204 | 0.084 | 0.204 | 0.111 | 0.217 | 1.030 | 0.842 | 0.117 | 0.227 | 0.069 | 0.174 | 0.120 | 0.235 | 1.036 | 0.843 | 0.106 | 0.219 |
| | 24 | **0.072** | 0.182 | 0.093 | 0.201 | 0.190 | 0.326 | 0.103 | 0.247 | 0.276 | 0.169 | 0.222 | 0.305 | 1.194 | 0.912 | 0.229 | 0.320 | 0.098 | 0.209 | 0.218 | 0.318 | 1.180 | 0.860 | 0.182 | 0.296 |
| | 36 | **0.125** | **0.247** | 0.159 | 0.260 | 0.256 | 0.376 | 0.263 | 0.391 | 0.245 | 0.390 | 0.365 | 0.396 | 1.351 | 0.975 | 0.370 | 0.414 | 0.165 | 0.276 | 0.329 | 0.400 | 1.303 | 0.909 | 0.261 | 0.366 |
| | 48 | **0.300** | **0.379** | 0.323 | 0.406 | 0.327 | 0.426 | 0.335 | 0.448 | 0.317 | 0.495 | 0.520 | 0.485 | 1.499 | 1.033 | 0.515 | 0.506 | 1.149 | 0.792 | 0.450 | 0.478 | 1.414 | 0.960 | 0.319 | 0.410 |
| | avg. | **0.138** | **0.239** | 0.160 | 0.262 | 0.226 | 0.351 | 0.199 | 0.323 | 0.231 | 0.315 | 0.305 | 0.351 | 1.269 | 0.941 | 0.308 | 0.367 | 0.370 | 0.363 | 0.279 | 0.358 | 1.233 | 0.893 | 0.217 | 0.323 |
| PEMS-BAY | 12 | **0.342** | **0.258** | 0.363 | 0.276 | 0.400 | 0.290 | 0.396 | 0.285 | 0.400 | 0.290 | 0.441 | 0.285 | 1.299 | 0.629 | 0.433 | 0.297 | 0.379 | 0.280 | 0.441 | 0.306 | 1.303 | 0.626 | 0.425 | 0.304 |
| | 24 | **0.607** | **0.350** | 0.633 | 0.374 | 0.647 | 0.382 | 0.659 | 0.388 | 0.640 | 0.383 | 0.717 | 0.380 | 1.390 | 0.644 | 0.699 | 0.393 | 0.733 | 0.440 | 0.706 | 0.399 | 1.395 | 0.645 | 0.673 | 0.409 |
| | 36 | **0.764** | **0.401** | 0.785 | 0.428 | 0.801 | 0.446 | 0.833 | 0.452 | 0.798 | 0.432 | 0.918 | 0.452 | 1.457 | 0.664 | 0.900 | 0.461 | 1.290 | 0.601 | 0.907 | 0.462 | 1.461 | 0.660 | 0.845 | 0.478 |
| | 48 | **0.917** | **0.465** | 0.951 | 0.497 | 0.955 | 0.493 | 0.951 | 0.493 | 1.291 | 0.601 | 1.072 | 0.504 | 0.988 | 1.809 | 1.050 | 0.509 | 1.291 | 0.601 | 1.042 | 0.502 | 1.510 | 0.673 | 0.959 | 0.520 |
| | avg. | **0.658** | **0.369** | 0.683 | 0.394 | 0.701 | 0.403 | 0.710 | 0.405 | 0.782 | 0.427 | 0.787 | 0.405 | 1.284 | 0.937 | 0.771 | 0.415 | 0.923 | 0.481 | 0.774 | 0.417 | 1.417 | 0.651 | 0.726 | 0.428 |
| METR-LA | 12 | **0.383** | **0.243** | 0.403 | 0.275 | 0.417 | 0.279 | 0.412 | 0.319 | 0.436 | 0.319 | 0.448 | 0.283 | 1.002 | 0.620 | 0.425 | 0.317 | 1.000 | 0.618 | 0.429 | 0.333 | 0.429 | 0.333 | 0.414 | 0.335 |
| | 24 | **0.583** | **0.359** | 0.600 | 0.366 | 0.603 | 0.362 | 0.622 | 0.413 | 0.647 | 0.417 | 0.677 | 0.381 | 1.089 | 0.652 | 0.637 | 0.418 | 1.000 | 0.606 | 0.635 | 0.427 | 1.090 | 0.645 | 0.605 | 0.446 |
| | 36 | **0.701** | **0.442** | 0.725 | 0.448 | 0.776 | 0.490 | 0.803 | 0.508 | 0.824 | 0.494 | 0.832 | 0.456 | 1.165 | 0.670 | 0.793 | 0.492 | 1.167 | 0.669 | 0.789 | 0.494 | 1.166 | 0.668 | 0.737 | 0.517 |
| | 48 | 0.826 | 0.513 | **0.823** | 0.507 | 0.833 | 0.505 | 0.861 | 0.539 | 0.951 | 0.548 | 0.949 | 0.516 | 1.234 | 0.690 | 0.913 | 0.540 | 1.176 | 0.654 | 0.910 | 0.541 | 1.234 | 0.689 | 0.834 | 0.571 |
| | avg. | **0.623** | **0.389** | 0.638 | 0.399 | 0.657 | 0.409 | 0.675 | 0.445 | 0.715 | 0.445 | 0.727 | 0.409 | 1.123 | 0.658 | 0.692 | 0.442 | 1.086 | 0.637 | 0.691 | 0.449 | 0.980 | 0.584 | 0.648 | 0.467 |

$$\text{MSE} = \frac{1}{T} \sum_{j=0}^{T-1} \|\mathbf{Y}_j - \hat{\mathbf{Y}}_j\|_2^2, \tag{25}$$

$$\text{MAE} = \frac{1}{T} \sum_{j=0}^{T-1} \left\| \mathbf{Y}_j - \hat{\mathbf{Y}}_j \right\|, \tag{26}$$

where $T$ denotes the number of data points (i.e., the forecasting horizon). $\mathbf{Y}_j$ and $\hat{\mathbf{Y}}_j$ are the ground truth and predicted values of the $j$-th data point, respectively.

# E. Full Results

We compare TimeMRA with 10 baselines across three different tasks: long-term time series forecasting, short-term time series forecasting, and few-shot time series forecasting. For a fair comparison, we evaluate TimeMRA and baselines under unified settings of each task. The average results refer to the mean of results under different forecasting horizons, where the best results are **bolded** and the second best results are underlined.

**Long-Term Time Series Forecasting**. Table 9 summarizes the results of long-term time series forecasting. * indicates that some baselines do not adhere to the unified settings or do not have results on the specific dataset. Therefore, we reran their official code and fine-tuned the key hyperparameters. Other results are from TimeCMA (Liu et al., 2025). From Table 9, we can observe that TimeMRA achieves the SOTA results on all datasets. Specifically, TimeMRA achieves average error reductions of 9.44% and 6.71% compared to the latest alignment-generative LLM, S²IP-LLM, in terms of MES and MAE, respectively.

**Short-Term Time Series Forecasting**. Table 10 summarizes the full results of short-term time series forecasting. We can observe that TimeMRA achieves the SOTA results on almost all datasets. Specifically, TimeMRA gives an average error reduction of 11.73% and 10.47% compared to other retrieval-augmented LLMs (i.e., TimeCMA and UniTime) in terms of MSE and MAE, respectively, and surpasses alignment-generative LLMs (i.e., S²IP-LLM and FPT) by an average of 19.10% and 16.24% in terms of MSE and MAE, respectively.

**Few-Shot Time Series Forecasting**. LLM4TS methods demonstrate strong robustness even under extreme conditions (e.g., few-shot scenarios). We use 10% and 5% of the training data from the ETT dataset to evaluate the few-shot learning performance. Table 11 and Table 12 summarize the full results of few-shot learning under 10% and 5% training data,

*Table 11.* Full results of few-shot time series forecasting under 10% training data. The input length is set to 96, and the forecasting lengths are set to $\{96, 192, 336, 720\}$. Lower MSE and MAE indicate better performance. Best results are in **bold** and the second best are underlined.

| Methods | | TimeMRA | | TimeCMA | | UniTime | | S²IP-LLM | | FPT | |
|---|---|---|---|---|---|---|---|---|---|---|---|
| Metric | | MSE | MAE | MSE | MAE | MSE | MAE | MSE | MAE | MSE | MAE |
| ETTh1 | 96 | **0.447** | **0.409** | 0.462 | 0.432 | 0.674 | 0.535 | 0.669 | 0.550 | 0.474 | 0.455 |
| | 192 | **0.498** | **0.475** | 0.532 | 0.482 | 0.687 | 0.551 | 0.692 | 0.562 | 0.551 | 0.495 |
| | 336 | **0.614** | **0.523** | 0.639 | 0.565 | 0.701 | 0.566 | 0.724 | 0.558 | 0.635 | 0.542 |
| | 720 | **0.700** | **0.576** | 0.722 | 0.632 | 0.716 | 0.594 | 0.896 | 0.667 | 0.755 | 0.611 |
| | avg. | **0.565** | **0.496** | 0.589 | 0.528 | 0.695 | 0.562 | 0.745 | 0.584 | 0.604 | 0.526 |
| ETTh2 | 96 | **0.317** | **0.349** | 0.358 | 0.390 | 0.340 | 0.380 | 0.347 | 0.382 | 0.333 | 0.364 |
| | 192 | **0.376** | **0.386** | 0.448 | 0.441 | 0.425 | 0.425 | 0.458 | 0.444 | 0.395 | 0.400 |
| | 336 | **0.435** | **0.442** | 0.513 | 0.486 | 0.454 | 0.450 | 0.568 | 0.510 | 0.569 | 0.498 |
| | 720 | **0.431** | **0.445** | 0.516 | 0.497 | 0.456 | 0.461 | 0.641 | 0.558 | 1.049 | 0.740 |
| | avg. | **0.390** | **0.406** | 0.459 | 0.454 | 0.419 | 0.429 | 0.504 | 0.474 | 0.587 | 0.501 |
| ETTm1 | 96 | **0.473** | **0.432** | 0.583 | 0.490 | 0.490 | 0.444 | 0.582 | 0.490 | 0.611 | 0.488 |
| | 192 | **0.505** | **0.438** | 0.554 | 0.485 | 0.507 | 0.454 | 0.552 | 0.476 | 0.571 | 0.483 |
| | 336 | **0.511** | **0.452** | 0.574 | 0.501 | 0.538 | 0.470 | 0.575 | 0.497 | 0.610 | 0.509 |
| | 720 | **0.571** | **0.474** | 0.726 | 0.569 | 0.590 | 0.498 | 0.603 | 0.512 | 0.625 | 0.516 |
| | avg. | **0.515** | **0.449** | 0.609 | 0.511 | 0.531 | 0.467 | 0.578 | 0.494 | 0.604 | 0.499 |
| ETTm2 | 96 | **0.160** | **0.255** | 0.207 | 0.282 | 0.199 | 0.276 | 0.191 | 0.269 | 0.183 | 0.263 |
| | 192 | 0.258 | 0.316 | 0.266 | 0.317 | 0.260 | 0.312 | 0.255 | **0.309** | 0.253 | **0.309** |
| | 336 | **0.305** | **0.342** | 0.325 | 0.355 | 0.318 | 0.348 | 0.321 | 0.353 | 0.320 | 0.350 |
| | 720 | **0.383** | **0.381** | 0.439 | 0.423 | 0.417 | 0.404 | 0.443 | 0.422 | 0.454 | 0.426 |
| | avg. | **0.277** | **0.324** | 0.309 | 0.344 | 0.299 | 0.335 | 0.303 | 0.338 | 0.303 | 0.337 |

*Table 12.* Full results of few-shot time series forecasting under 5% training data. The input length is set to 96, and the forecasting lengths are set to $\{96, 192, 336, 720\}$. Lower MSE and MAE indicate better performance. Best results are in **bold** and the second best are underlined. $--$ indicates that 5% of the training data is insufficient to constitute a complete training set.

| Methods | | TimeMRA | | TimeCMA | | UniTime | | S²IP-LLM | | FPT | |
|---|---|---|---|---|---|---|---|---|---|---|---|
| Metric | | MSE | MAE | MSE | MAE | MSE | MAE | MSE | MAE | MSE | MAE |
| ETTh1 | 96 | **0.475** | **0.448** | 0.498 | 0.501 | 0.685 | 0.542 | 0.769 | 0.608 | 0.508 | 0.476 |
| | 192 | **0.694** | **0.532** | 0.784 | 0.669 | 0.719 | 0.558 | 0.937 | 0.687 | 0.725 | 0.588 |
| | 336 | **0.719** | **0.541** | 0.756 | 0.576 | 0.752 | 0.581 | 1.123 | 0.730 | 0.774 | 0.603 |
| | 720 | $--$ | $--$ | $--$ | $--$ | $--$ | $--$ | $--$ | $--$ | $--$ | $--$ |
| | avg. | **0.629** | **0.507** | 0.679 | 0.582 | 0.719 | 0.560 | 0.943 | 0.675 | 0.669 | 0.556 |
| ETTh2 | 96 | **0.314** | **0.322** | 0.331 | 0.357 | 0.341 | 0.380 | 0.376 | 0.402 | 0.338 | 0.369 |
| | 192 | **0.397** | **0.413** | 0.427 | 0.419 | 0.432 | 0.428 | 0.562 | 0.505 | 0.418 | 0.419 |
| | 336 | **0.459** | 0.463 | 0.489 | 0.465 | 0.478 | **0.461** | 0.629 | 0.545 | 0.828 | 0.642 |
| | 720 | $--$ | $--$ | $--$ | $--$ | $--$ | $--$ | $--$ | $--$ | $--$ | $--$ |
| | avg. | **0.390** | **0.399** | 0.416 | 0.414 | 0.417 | 0.423 | 0.522 | 0.484 | 0.528 | 0.477 |
| ETTm1 | 96 | **0.486** | **0.417** | 0.545 | 0.480 | 0.505 | 0.445 | 0.659 | 0.540 | 0.565 | 0.483 |
| | 192 | 0.563 | **0.455** | 0.623 | 0.519 | 0.573 | 0.476 | 0.628 | 0.529 | **0.560** | 0.486 |
| | 336 | **0.603** | **0.495** | 0.692 | 0.557 | 0.619 | 0.499 | 0.702 | 0.572 | 0.625 | 0.520 |
| | 720 | **0.711** | **0.539** | 0.726 | 0.551 | 0.732 | 0.551 | 0.892 | 0.647 | 0.831 | 0.609 |
| | avg. | **0.591** | **0.477** | 0.648 | 0.527 | 0.611 | 0.496 | 0.720 | 0.572 | 0.645 | 0.525 |
| ETTm2 | 96 | **0.175** | 0.280 | 0.221 | 0.302 | 0.194 | **0.271** | 0.207 | 0.290 | 0.198 | 0.282 |
| | 192 | **0.243** | **0.295** | 0.284 | 0.337 | 0.260 | 0.312 | 0.277 | 0.333 | 0.269 | 0.326 |
| | 336 | **0.309** | **0.328** | 0.338 | 0.368 | 0.327 | 0.353 | 0.333 | 0.364 | 0.317 | 0.347 |
| | 720 | **0.403** | **0.408** | 0.454 | 0.434 | 0.431 | 0.412 | 0.427 | 0.428 | 0.482 | 0.445 |
| | avg. | **0.283** | **0.328** | 0.324 | 0.360 | 0.303 | 0.337 | 0.311 | 0.354 | 0.317 | 0.350 |

respectively. From Table 11, we can observe that TimeMRA performs better than other LLM4TS methods under 10% training data settings. Specifically, TimeMRA gives an average error reduction of 9.01% and 6.09% compared to the second best baselines in terms of MSE and MAE, respectively.

In addition, from Table 12, we observe that TimeMRA still achieves the best results, even with fewer training data. The reason may be that the multi-modality encoder (MME) module can extract temporal features and disentangled multi-scale semantic representations that are beneficial for time series forecasting. Specifically, TimeMRA reduces forecasting errors by an average of 14.63% and 11.31% compared to other LLM4TS methods (i.e., TimeCMA, UniTime, S²IP-LLM, and FPT) in terms of MSE and MAE, respectively.

# F. Error Bars

All experimental results reported in the main text and appendix are averaged over three independent runs with different random seeds. To evaluate the robustness of TimeMRA to the choice of random seeds, we report the standard deviation

under long-term time series forecasting settings. The experimental results on different datasets are summarized in Table 13 and Table 14, respectively. We can observe that the variances are considerably small, indicating the robustness of TimeMRA against the choice of random seeds.

*Table 13.* The standard deviation results of TimeMRA on Weather, Electricity, and Traffic datasets. Results are averaged from three random seeds.

| Dataset | Weather | Electricity | Exchange |
|---|---|---|---|
| Horizon | MSE MAE | MSE MAE | MSE MAE |
| 96 | 0.159±0.0007 0.187±0.00012 | 0.129±0.0005 0.233±0.0008 | 0.081±0.0001 0.198±0.0002 |
| 192 | 0.216±0.0012 0.261±0.00007 | 0.155±0.0009 0.241±0.0002 | 0.172±0.0003 0.295±0.0004 |
| 336 | 0.247±0.0013 0.276±0.0009 | 0.157±0.0003 0.263±0.0003 | 0.305±0.0004 0.389±0.0003 |
| 720 | 0.334±0.0006 0.317±0.0003 | 0.224±0.0004 0.298±0.0001 | 0.831±0.0006 0.681±0.0009 |

*Table 14.* The standard deviation results of TimeMRA on ETT dataset. Results are averaged from three random seeds.

| Dataset | ETTh1 | ETTh2 | ETTm1 | ETTm2 |
|---|---|---|---|---|
| Horizon | MSE MAE | MSE MAE | MSE MAE | MSE MAE |
| 96 | 0.369±0.0009 0.375±0.0003 | 0.275±0.0008 0.325±0.0004 | 0.303±0.0015 0.347±0.007 | 0.163±0.0005 0.253±0.0004 |
| 192 | 0.428±0.0010 0.423±0.0005 | 0.355±0.0006 0.371±0.0003 | 0.352±0.0011 0.364±0.0013 | 0.231±0.0003 0.295±0.0007 |
| 336 | 0.426±0.0013 0.431±0.0004 | 0.414±0.0007 0.427±0.0002 | 0.365±0.0044 0.382±0.0022 | 0.275±0.0009 0.331±0.0005 |
| 720 | 0.443±0.0001 0.447±0.0006 | 0.417±0.0011 0.435±0.0003 | 0.440±0.0077 0.431±0.0054 | 0.379±0.0011 0.385±0.0003 |

