# OpenReview forum: "TimeMRA: LLM-Empowered Time Series Forecasting via Multi-Scale Retrieval-Augmented Representations"
_ICML.cc/2026/Conference — ICML 2026 regular_

### Official Review · Reviewer_s7go · 2026-03-06

**Soundness:** 4
**Presentation:** 4
**Significance:** 4
**Originality:** 3
**Overall Recommendation:** 5
**Confidence:** 5

**Summary:**

The authors propose TimeMRA, a framework designed to improve LLM-based time series forecasting by addressing the issues of entangled multi-scale representations and redundant multi-scale interference. Specifically, a scale-aware prompt generation module is designed to decompose time series into multiple scales into generate corresponding prompts and a router network is designed to select multi-scale representations. In addition, a cross-scale disentanglement constraint mechanism is used to perform cross-scale contrastive learning. Experimental results verify the effectiveness of the proposed method.

**Compliance With Llm Reviewing Policy:**

Affirmed.

**Final Justification:**

All my previous concerns are addressed. Overall, I think this paper is clearly written, well-motivated, and empirically valid.

**Key Questions For Authors:**

1. What is the impact of adjusting the LLM size on the frozen LLM used to generate multi-scale representations? How does the model's performance change with different LLM sizes or different layer configurations within the same LLM?

2. What is the role of the Router Network? Is it necessary to add ablation experiments?

3. How robust is TimeMRA under extreme conditions?

**Limitations:**

The authors didn't discuss limitations.

**Strengths And Weaknesses:**

## Strengths

1. The concept of multi-scale retrieval-augmented representations is novel and compelling, and the design of CSDC mechanism via CLIP-style contrastive learning is a well-grounded approach to solving the scale entanglement problem in multi-scale modeling.

2. This paper is well-organized and easy to follow.

3. The experimental results and ablation studies are solid, and the comprehensive experiments conducted on 10 datasets demonstrate consistent SOTA performance.

## Weaknesses

1. This paper has limited research on scaling LLM models, which limits whether the scaling law applies to frozen LLM models that generate multi-scale representations.

2. This paper's explanation of the role of the Router Network is unclear, and corresponding ablation experiments are lacking.

3. This paper has limited research on the robustness of the model under extreme conditions, especially in scenarios involving anomaly injection and ultra-long-term predictions.

---

> ### Author Rebuttal · Authors · 2026-03-31
>
> We sincerely thank the reviewer for recognizing our concept as novel and compelling, and for acknowledging our method as a well-grounded approach. For the thoughtful and constructive comments, we carefully address each concern below.
>
> **Q1:** How do LLM size and layer selection affect the quality of multi-scale representations and forecasting performance?
>
> Thanks for your valuable comments and scientific rigor. Scaling law is an essential characteristic that **extends from small models to large foundation models**. Experimental results in Table 5 of the original paper indicate that large LLMs enable TimeMRA to produce more informative and expressive multi-scale augmented representations. To further investigate the impact of different LLM layers used to generate multi-scale representations, we design the following variants:
>
> -G.4: Using the first 4 Transformer layers of GPT-2 Small.
>
> -G.8: Using the first 8 Transformer layers of GPT-2 Small.
>
> Table 1
>
> |Methods|-G.4|-G.8|TimeMRA|
> |---|---|---|---|
> |Metric|MSE MAE|MSE MAE|MSE MAE|
> |96|0.377 0.387|0.370 0.379|**0.369 0.375** |
> |192|0.430 0.431|0.434 **0.423**| **0.428 0.423** |
> |336|0.447 0.453|0.436 0.442|**0.426 0.431**|
>
> The experimental results on ETTh1 dataset (Table 1) show that TimeMRA performs better than G.4 and G.8 in almost all cases, which indicates that **the scaling law also applies to multi-scale augmented representations with frozen LLMs**.
>
> **Q2:** What is the role of the Router Network? Is it necessary to add ablation experiments?
>
> The router network is designed to adaptively **select critical multi-scale representations while mitigating interference from irrelevant scales**. To investigate the effectiveness of the router network, we design the following variants:
>
> -w/o Router: Removing the router network and using all scales equally for cross-modality retrieval.
>
> -LW: Replacing the router network with learnable scale-wise weights that aggregate multi-scale representations via weighted summation.
>
> Table 2
>
> |Methods|-w/o Router|-LW|TimeMRA|
> |---|---|---|---|
> |Metric| MSE MAE | MSE MAE | MSE MAE |
> |96| 0.375 0.384 | 0.372 0.379 | **0.369 0.375** |
> |192| 0.435 0.430 | 0.438 0.425 | **0.428 0.423** |
> |336| 0.443 0.441 | 0.431 0.437 | **0.426 0.431** |
>
> The experimental results on ETTh1 dataset (Table 2) show that: (1) TimeMRA performs better than -LW in all cases, showing the effectiveness of the router network. (2) -w/o Router gets the worst performance. The reason is that **treating all scales equally may introduce redundant information from less relevant scales**.
>
> **Q3:** How robust is TimeMRA under extreme conditions?
>
> To evaluate the robustness of TimeMRA, we compare it with LLM4TS methods under anomaly injection and ultra-long-term forecasting conditions. The corresponding results are presented below:
>
> **Anomaly Injection.** We conduct experiments by injecting randomly generated anomalies in the training data. The anomaly rate varies from 10% to 20%. The experiments are conducted on ETTh1 dataset with the input length I=96 and output length O=96.
>
> Table 3
>
> |Methods |TimeMRA|TimeCMA|UniTime|S$^2$IP-LLM|
> |---|---|---|---|---|
> |Metric | MSE MAE |MSE MAE|MSE MAE|MSE MAE|
> |0%  | **0.369 0.375**|0.373 0.391|0.397 0.418 |0.390 0.400|
> |10% | **0.377 0.384**|0.392 0.496|0.415 0.503 |0.511 0.604|
> | 15% | **0.433 0.435**|0.613 0.568|0.443 0.572 |0.733 0.584|
> |20% | **0.741 0.590**|0.780 0.599|0.809 0.613|0.817 0.610|
>
> The experimental results are summarized in Table 3. We observe that TimeMRA achieves the best performance in all cases, showing its superior ability **in time series forecasting even under scenarios with anomaly injection**. The reason is that the multi-scale decomposition in the SAPG module performs frequency analysis on the entire training data rather than individual input sequences, **making the dominant period selection less sensitive to local anomalies**. In addition, the router network **adaptively selects informative scales and suppresses redundant multi-scale interference**, which also enhances the robustness of TimeMRA against anomaly injection.
>
> **Ultra-Long-Term Forecasting.** We conduct ultra-long-term forecasting on ETTh1 dataset by taking a fixed input length (I=96) to predict ultra-long horizons (O={1008, 1440, 1800}).
>
> Table 4
>
> |Methods|TimeMRA|TimeCMA|UniTime|S$^2$IP-LLM|
> |--- |---|---|---|---|
> |Metric|MSE MAE|MSE MAE|MSE MAE|MSE MAE|
> |1008| **0.512 0.499**|0.535 0.517|0.557 0.511|0.564 0.526|
> |1440| **0.586 0.477**|0.611 0.495|0.631 0.517|0.644 0.527|
> |1800| **0.665 0.573**|0.712 0.622|0.733 0.625|0.764 0.641|
>
> The experimental results are summarized in Table 4. We observe that TimeMRA achieves SOTA results on almost all cases, demonstrating its effectiveness for ultra-long-term forecasting. The reason may be that the SAPG module **performs frequency analysis on the entire training dataset**, thereby capturing **dominant multi-scale temporal patterns** that are essential for ultra-long-term forecasting.

---

> > ### Author Rebuttal · Reviewer_s7go · 2026-04-02
> >
> > Thanks for the detailed rebuttal, all my concerns are well-addressed.

---

> > > ### Author Response · Authors · 2026-04-03
> > >
> > > We're glad to have fully remedied your concerns! Thank you very much for engaging and for raising your score. We sincerely appreciate the time and effort you dedicated to reviewing our manuscript. Your insightful comments and suggestions were invaluable in helping us improve the quality of this work.

---

### Official Review · Reviewer_b1S9 · 2026-03-08

**Soundness:** 2
**Presentation:** 3
**Significance:** 2
**Originality:** 3
**Overall Recommendation:** 3
**Confidence:** 5

**Summary:**

The paper proposes TimeMRA, a multi-scale retrieval-augmented framework for LLM-empowered time-series forecasting that aims to reduce cross-scale entanglement and suppress interference from irrelevant scales. TimeMRA first decomposes the input into multiple scales and uses a frozen LLM to generate scale-aware semantic embeddings. It then applies a cross-scale disentanglement constraint and a router to select the most informative scales before prediction. Experiments on 10 real-world datasets report that TimeMRA achieves state-of-the-art forecasting performance relative to a broad set of classical and LLM-based baselines.

**Compliance With Llm Reviewing Policy:**

Affirmed.

**Ethical Review Concerns:**

I first assumed the paper uses prompt injection. However, I believe this was added by the PCs.

**Ethical Review Flag:**

Flag this paper for an ethics review.

**Key Questions For Authors:**

I have the following questions for the authors.
1. How does the approach constitute "retrieval" in any meaningful sense? There is no external database, no stored index, and no query-time lookup. Can the authors clarify what distinguishes CMR from standard cross-modal attention fusion, and why the term "retrieval-augmented" is appropriate here?
2. Table 5 shows that stronger LLM backbones (LLaMA2-7B, Qwen2.5-7B, DeepSeekR1-8B) consistently outperform GPT2-small, yet GPT2-small is used as the default throughout all main experiments. Why was the weakest backbone chosen as default LLM backbone?
3. All experiments fix input length at I=96. Given that the multi-scale decomposition relies on FFT-based period detection, do the identified dominant periods carry meaningful spectral content at this short length?

**Limitations:**

- All experiments fix the input length at 96, which is unusually short. It is unclear whether the multi-scale decomposition provides meaningful periodic structure at this length, and whether results would hold for longer look-back windows common in practice.
- The LLM is fully frozen throughout training, meaning the semantic representations it produces are generic text embeddings not adapted to time series semantics. However, I noticed the paper does not analyze whether fine-tuning even a subset of LLM parameters would yield different conclusions about which components matter most.
- The method produces only point forecasts with MSE/MAE evaluation. For many of the application domains cited (healthcare, finance, energy), calibrated probabilistic forecasts are essential, and the paper provides no path toward this. The authors may want to try WQL as well.
- Limited related works as there is no discussion of retrieval-augmented time series forecasting methods.

**Strengths And Weaknesses:**

**Strengths**
-  The two identified failure modes of existing retrieval-augmented LLMs — entangled multi-scale representations and redundant multi-scale interference — are well-articulated and the proposed modules map directly onto these problems.
- Experiments span 10 datasets across long-term, short-term, and few-shot forecasting with good improvements on few-shot forecasting.
- The ablation study systematically isolates each component across multiple horizons, giving a clear picture of each contribution.
- The paper is well-organized.

**Weaknesses**
- I believe the "retrieval" framing is misleading. Despite being positioned as a retrieval-augmented method, the cross-modality retrieval module (Eq. 20) is simply a cross-attention operation between temporal query features and LLM-generated semantic key/value representations from the same input series. There is no actual retrieval from an external database or historical pool, which is fundamentally different from what retrieval-augmented typically implies in the literature.
- Modern strong non-LLM baselines such as TimeMixer, iTransformer, and more recent TSFMs like TimesFM or Moirai are absent or only partially included.
- Despite using a frozen LLM at inference time across S scales and N variables, no wall-clock time or memory comparison against baselines is provided. The complexity analysis in Appendix G acknowledges LLM inference cost but does not quantify it empirically, which is I believe a significant omission for a method that processes multiple scales through a large language model.
- The paper omits several directly relevant works from the retrieval-augmented time series forecasting literature. (RAF; https://arxiv.org/abs/2411.08249, RAFT; https://openreview.net/forum?id=GYwH71ugtC, TS-RAG; https://arxiv.org/abs/2503.07649)

---

> ### Author Rebuttal · Authors · 2026-03-31
>
> We thank the reviewer for the valuable feedback, especially recognizing that our motivation is well-articulated, the paper is well-organized, and the ablation studies give a clear picture of each contribution. For the constructive comments, we address each concern below.
>
> **W1&Q1:** Explain why retrieval-augmented is appropriate and clarify what distinguishes CMR from standard cross-modal attention fusion (SCAF)?
>
> Thanks for your valuable feedback and scientific rigor. Unlike traditional RAG systems that query a static external database, TimeMRA constructs a sample-specific, multi-scale dynamic knowledge base by **treating the LLM-generated multi-scale representations as a retrieval source**. However, TimeMRA can also construct an external database in a static manner. To further investigate this, we design the following variant:
>
> -avg: Replacing the learnable aggregation layer with average pooling to extract static multi-scale features, which are precomputed and **stored as a fixed knowledge base** for cross-modality retrieval.
>
> Table 1
>
> |Methods|-avg|TimeMRA|
> |---|---|---|
> |Metric|MSE MAE|MSE MAE|
> |96|**0.369** 0.383 |**0.369 0.375**|
> |192|0.435 0.437|**0.428 0.423**|
> |336|0.439 0.441|**0.426 0.431**|
>
> The results on ETTh1 dataset (Table 1) show that TimeMRA outperforms -avg in all cases. The reason is that TimeMRA extract augmented multi-scale features in a learnable manner, which is **more flexible and can capture more complex temporal patterns**.
>
> The key distinctions between CMR and SCAF are as follows: (1) SCAF operates on a fixed pair of representations, whereas CMR retrieves from a multi-scale knowledge base with $S$ candidate entries per variable. (2) SCAF treats all keys equally, whereas CMR **explicitly prunes irrelevant candidates** through the router network before retrieval. (3) The knowledge base in CMR is dynamically constructed by LLMs conditioned on the input, **providing richer and more adaptive semantic context than static fusion**.
>
> **W2&3&4:** Add more baselines and complexity analysis.
>
> **Compared with non-LLM baselines**: TimeMixer and iTransformer have been included for long/short-term forecasting. To further investigate the effectiveness of TimeMRA over non-LLM baselines, we evaluate the zero-shot performance on ETT dataset.
>
> Table 2
>
> |Methods|5% training data|10% training data|
> |---|---|---|
> |TimeMixer|0.654 0.537|0.603 0.565|
> | iTransformer|0.675 0.542|0.616 0.570|
> |TimeCMA|0.517 0.471|0.492 0.459|
> |TimeMRA|**0.473 0.428**|**0.437 0.419**|
>
> The results in Table 2 show that LLM4TS methods outperform non-LLM baselines by a large margin. The reason is that non-LLM baselines, trained from scratch, suffer from limited training data under this scenario. In contrast, LLM4TS methods can **leverage pre-trained knowledge and align it with time series representations**, thereby enhancing their ability for time series forecasting.
>
> **Compared with TSFMs and retrieval-augmented methods**: Following the reviewer's suggestions, we have added Moirai, Chronos, RAFT, and TS-RAG for comparison.
>
> Table 3
>
> |Methods|Weather|ECL|Exchange|ETT|
> |---|---|---|---|---|
> |Moirai|0.283 0.305|0.223 0.312|0.375 0.421| 0.382 0.417|
> |Chronos |0.296 0.297|0.214 0.307|0.382 0.415| 0.393 0.419|
> |RAFT|0.270 0.309|0.179 0.282|0.382 0.408|0.371 0.397|
> |TS-RAG|0.264 0.283| 0.202 0.289|0.369 0.408|0.373 0.392|
> |TimeMRA|**0.239 0.260**|**0.166 0.259**|**0.347 0.391**|**0.352 0.377**|
>
> The results in Table 3 show that: (1) TimeMRA performs better than TSFMs (i.e., Moirai and Chronos). The reason is that LLMs are trained on large-scale datasets, which **endow them with strong feature extraction and generalization abilities**. (2) TimeMRA performs better than other retrieval-augmented methods in all datasets. This is because by leveraging the MME module and cross-modality retrieval module, TimeMRA can **obtain multi-scale retrieval-augmented representations that are beneficial for forecasting**.
>
> We have added both LLM4TS methods and non-LLM baselines to evaluate the computational cost. Due to space constraints, please refer to our response to **Reviewer e1vf** for details.
>
> **Q2:** Why was the weakest backbone chosen as default LLM backbone?
>
> For a fair comparison, following existing methods (e.g., TimeCMA, UniTime, and FPT), we adopt GTP2-small as the default LLM backbone.
>
> **Q3:** Do the identified dominant periods carry meaningful spectral content at fix input length I=96?
>
> We would like to clarify that the FFT-based period detection **is performed on the entire training data** (as described in Section 4.1) to obtain the dominant periodic patterns. Then, **spectral projection** is applied to align the high-resolution amplitude spectrum with the input sequence length. Ablation studies in Section 5.2 show that replacing the FFT-based multi-scale decomposition with predefined aggregation windows (-PAW) leads to performance degradation, confirming that **the identified dominant periods are informative and dataset-adaptive**.

---

> > ### Author Rebuttal · Reviewer_b1S9 · 2026-04-03
> >
> > Some of my concerns are resolved.
> >
> > (i) There were no results regarding the existing retrieval method RAF (AISTATS, 2026).
> >
> > (ii) I could not see any results about probabilistic forecasting.

---

> > > ### Author Response · Authors · 2026-04-04
> > >
> > > Thanks for your valuable feedback and scientific rigor. Actually, we have carefully taken all of your comments into consideration, and we apologize for omitting some baselines and evaluation metrics for comparison in our first-round response due to space limitations. The full results are presented as follows:
> > >
> > > **1** Add RAF for comparison.
> > >
> > > Thanks for your comments. We have added RAF for comparison. The averaged experimental results on Weather, ECL, Exchange, and ETT (ETTh1, ETTh2, ETTm1, and ETTm2) datasets are shown in Table 1.
> > >
> > > Table 1
> > >
> > > | Methods | Weather | ECL | Exchange | ETT |
> > > | --- | --- | --- | --- | --- |
> > > | Metric | MSE MAE | MSE MAE | MSE MAE | MSE MAE |
> > > | RAF | 0.267 0.286 | 0.205 0.293 | 0.374 0.411 | 0.378 0.404 |
> > > | TimeMRA | **0.239 0.260** | **0.166 0.259** | **0.347 0.391** | **0.352 0.377** |
> > >
> > > From Table 1, we can observe that TimeMRA performs better than RAF in all cases. This is because **by leveraging the MME module and cross-modality retrieval module**, TimeMRA can obtain **multi-scale retrieval-augmented representations that are beneficial for forecasting**. We have incorporated the full results in the revised paper and added descriptions of corresponding retrieval-augmented methods in the **Related Work section**.
> > >
> > > **2** Add WQL to evaluate the probabilistic forecasting.
> > >
> > > To evaluate the probabilistic forecasting performance of TimeMRA over existing methods, we replace the final linear projection layer with a multi-quantile output head that simultaneously predicts 9 quantile levels ($\tau \in \{0.1, 0.2, 0.3, 0.4, 0.5, 0.6, 0.7, 0.8, 0.9\}$). The MSE loss is replaced with the quantile loss. The experimental results on the WQL metric are shown in Table 2.
> > >
> > > Table 2
> > >
> > > | Methods | Weather | ECL | Exchange | ETT |
> > > | --- | --- | --- | --- | --- |
> > > | iTransformer | 0.235 | 0.224 | 0.339 | 0.286 |
> > > | PatchTST | 0.241 | 0.231 | 0.335 | 0.294 |
> > > | FPT | 0.204 | 0.209 | 0.314 | 0.278 |
> > > | S$^2$IP-LLM | 0.199 | 0.213 | 0.307 | 0.279 |
> > > | UniTime | 0.191 | 0.205 | 0.279 | 0.272 |
> > > | TimeCMA | 0.186 | 0.182 | 0.291 | 0.265 |
> > > | TimeMRA | **0.172** | **0.171** | **0.267** | **0.251** |
> > >
> > > From Table 2, we can observe that: (1) **LLM4TS methods perform better than non-LLM baselines**. This can be attributed to the rich semantic representations extracted by LLMs, which encode not only point-wise predictive patterns but also implicit uncertainty patterns that are beneficial for probabilistic forecasting. (2) **TimeMRA consistently achieves the best WQL performance** across all datasets and forecasting horizons. Specifically, TimeMRA reduces WQL by an average of **6.12%** compared to the second-best baseline, TimeCMA. This suggests that the multi-scale retrieval-augmented representations not only improve point forecasting accuracy but also enhance the quality of uncertainty estimation.
> > >
> > > **3** Explain the reason for using frozen LLM and perform partial fine-tuning analysis.
> > >
> > > **Rationale for freezing the LLM.** Following established practices (e.g., TimeCMA), we adopt the frozen LLM backbone. Although the LLM backbone is frozen, we **introduce the CSDC mechanism to bridge the gap between generated LLM representations and time series semantics** through cross-scale contrastive alignment. Ablation studies in Section 5.2 show that removing the CSDC mechanism causes performance degradation, confirming its effectiveness in adapting the frozen LLM's output to time series semantics.
> > >
> > > **Partial fine-tuning analysis.** To address the reviewer's suggestion, we design the following variant:
> > >
> > > -LoRA: it fine-tunes positional embeddings and layer normalization layers using the low-rank adaptation technique.
> > >
> > > Table 3
> > >
> > > | Methods | -LoRA | TimeMRA |
> > > | --- | --- | --- |
> > > | Metric | MSE MAE | MSE MAE |
> > > | 96  | **0.367 0.373** | 0.369 0.375 |
> > > | 192 | 0.435 0.427 | **0.428 0.423** |
> > > | 336 | 0.441 0.450 | **0.426 0.431** |
> > >
> > > The results on ETTh1 dataset are shown in Table 3. We can observe that TimeMRA outperforms -LoRA in most cases, likely because **fine-tuning the LLM on noisy time series data may cause semantic hallucinations and compromise the generality of the pre-trained representations**. The experimental results verify the effectiveness of our frozen LLM design.
> > >
> > > Thanks again for your valuable feedback. All the revisions have been included in the revised paper. We sincerely hope our explanations align with your expectations and would be deeply grateful for your further recognition and support.

---

### Official Review · Reviewer_e1vf · 2026-03-12

**Soundness:** 2
**Presentation:** 3
**Significance:** 2
**Originality:** 2
**Overall Recommendation:** 3
**Confidence:** 2

**Summary:**

This paper introduces TimeMRA, a framework for time series forecasting that leverages Large Language Models (LLMs) to generate multi-scale retrieval-augmented representations. The authors identify two primary challenges in existing LLM-based forecasting: entangled multi-scale representations and redundant scale interference. To address these, they propose a scale-aware prompt generation (SAPG) module based on training-set-wide FFT analysis, a cross-scale disentanglement constraint (CSDC) using contrastive learning, and a router network for adaptive scale selection. The system is evaluated against several deep learning and LLM-based baselines across 10 datasets.

**Compliance With Llm Reviewing Policy:**

Affirmed.

**Key Questions For Authors:**

Can you provide a "Latency vs. Accuracy" table comparing TimeMRA to PatchTST and TimeCMA? The reviewer needs to see the cost of the added LLM inference layers.

What happens if the LLM backbone is replaced by a randomly initialized Transformer of the same depth? This would clarify if the "pre-trained semantic knowledge" is actually the driver of performance.

How does the model handle distribution shifts or new periodicities that appear after the initial FFT analysis of the training data?

**Limitations:**

The authors briefly mention reducing computational burden by freezing the LLM, but they do not adequately discuss the scalability limits of their multi-scale approach as the number of variables $N$ increases.

**Strengths And Weaknesses:**

The paper addresses an important problem in time series analysis: the inherent multi-periodicity of real-world data. From a systems and methodology standpoint, the use of spectral projection to resolve frequency mismatches between global training data and local input windows is a thoughtful detail. The empirical results show consistent improvements over several recent baselines in MSE and MAE metrics.

However, the research motivation and system design raise significant concerns regarding "over-engineering." The core of the performance gain appears to stem from multi-scale decomposition—a technique well-established in classical signal processing—yet it is wrapped in an extremely heavy LLM-augmented encoder pathway. The authors freeze the LLM to reduce costs, but the system still requires multiple tokenization and inference passes for $S$ different scales, where the optimal $S$ is found to be 6. This introduces substantial computational overhead compared to traditional deep learning models like PatchTST or DLinear without a clear analysis of the accuracy-versus-latency trade-off. Furthermore, the disentanglement mechanism (CSDC) is evaluated through t-SNE, which is useful for visualization but does not substitute for a rigorous ablation study showing that the LLM is actually capturing "semantics" that a smaller, specialized encoder could not. The reliance on a Top-S selection from the entire training set also raises questions about the system's adaptability to concept drift in online environments.

---

> ### Author Rebuttal · Authors · 2026-03-31
>
> We thank the reviewer for recognizing our key motivation, i.e., addressing the inherent multi-periodicity of real-world time series data. For the constructive comments, we address each concern below.
>
> **Q1:** Provide a Latency vs. Accuracy comparison to assess the computational cost of the added LLM inference layers.
>
> Thanks for your valuable feedback and scientific rigor. To provide a more comprehensive evaluation of the computation cost, we compare the inference time (s), parameter number, GPU Occupation (MB), and forecasting performance between LLM4TS methods (i.e., TimeMRA, TimeCMA, and S$^2$IP-LLM) and non-LLM baselines (i.e., iTransformer and PatchTST). The experimental results on ETTh1 dataset with I=96 and O=96 are shown in Table 1.
>
> Table 1
>
> |Methods |Inference Time|Parameters|GPU Occupation|MAE|
> |---|---|---|---|---|
> |TimeMRA|48.236|9,371,453|937|**0.375**|
> |TimeCMA |160.779 |12,497,220|1,121|0.391|
> |S$^2$IP-LLM |69.647|51,495,344| 3,643 |0.400|
> |iTransformer|7.849|841,568 |713|0.405|
> |PatchTST|**0.855**|**35,171** |**589**|0.413|
>
> From Table 1, we can observe that PatchTST runs faster in these methods, but it gets the worst forecasting results. Compared with LLM4TS methods, TimeMRA runs faster and gets the best forecasting results. Overall, **considering both the forecasting performance and the computation cost, TimeMRA demonstrates its superiority over existing methods**.
>
> **Q2:** Replacing LLM backbone with a randomly initialized Transformer of the same depth. This would clarify if the "pre-trained semantic knowledge" is actually the driver of performance.
>
> To isolate the contribution of pre-trained knowledge, we design the following variants:
>
> -RanInit: Replacing LLM with the randomly initialized Transformer of identical architecture and depth.
>
> -RLLM: Replacing LLM with simple linear layers (Table 12 of the original paper).
>
> Table 2
>
> |Methods|-RanInit|-RLLM|TimeMRA|
> |---|---|---|---|
> |Metric|MSE MAE |MSE MAE|MSE MAE|
> |96|0.371 0.379|0.400 0.393|**0.369 0.375**|
> |192|0.433 0.431|0.448 0.457|**0.428 0.423**|
> |336|0.437 0.440|0.486 0.466|**0.426 0.431**|
>
> The results on ETTh1 dataset (Table 2) show that: (1) Although **sharing the same architecture**, TimeMRA performs better than -RanInit, which confirms that **the pre-trained semantic knowledge is a critical factor influencing model performance**. (2) -RanInit performs better than -RLLM, indicating that the Transformer architecture itself **provides some benefit through its attention mechanism for contextual feature extraction**.
>
> **Q3:** How does the model handle distribution shifts or new periodicities that appear after the initial FFT analysis of the training data?
>
> FFT is applied to the training data rather than individual samples, enabling more **robust extraction of periodic patterns** with **reduced sensitivity** to local noise. In addition, the router network **adaptively selects relevant scales while suppressing irrelevant ones**, alleviating distribution shift to some extent. Few-shot forecasting experiments further demonstrate the robustness of our model under limited/distribution-constrained data.
>
> To further evaluate the model's resilience to distribution shifts, we assess the zero-shot forecasting performance of TimeMRA and other LLM4TS methods on the M3 and M4 datasets.
>
> Table 3
>
> |Methods|TimeMRA|TimeCMA|UniTime|S$^2$IP-LLM|FPT|
> |---|---|---|---|---|---|
> |M4→M3|**12.531**|12.713|12.964|13.015|13.060|
> |M3→M4|**12.993**|13.122|13.137|13.210|13.125|
>
> Table 3 shows the zero-shot learning results in terms of averaged SMAPE. It is notable that both M3 and M4 datasets **contain complex multi-scale temporal patterns and show different data distributions**. TimeMRA still achieves the best performance, which may be due to the SAPG module providing a **comprehensive set of candidate periodicities**, while the router network can **adaptively select the most informative scales for unseen distributions**.
>
> **L1:** Discuss the scalability limits of the multi-scale approach as the number of variables $N$ increases.
>
> As discussed in Appendix G, the time complexity of TimeMRA is primarily bounded by O($N^2$). Although we adopt a multi-scale design, the number of selected scales $S$ is a small constant (e.g., $S$=6), and the router network further limits computation to $M$ scales per variable (e.g., $M$=3). Consequently, the **computational overhead of our multi-scale design is marginal**. In addition, our experiments cover datasets with diverse variable scales, **ranging from small (Exchange, $N$=8) to large (PEMS03, $N$=358; Electricity, $N$=321)**. TimeMRA achieves SOTA results across all these settings, demonstrating that the multi-scale design does not introduce prohibitive overhead as $N$ increases. However, as pointed out by the reviewer, for very high-dimensional settings (e.g., $N$ > 1000), the O($N^2$) attention cost may become a bottleneck. Considering the scope of this paper, we would like to leave the exploration in future work.

---

### Official Review · Reviewer_C7pf · 2026-03-13

**Soundness:** 3
**Presentation:** 3
**Significance:** 3
**Originality:** 3
**Overall Recommendation:** 4
**Confidence:** 4

**Summary:**

This paper proposes TimeMRA, an LLM-empowered time series forecasting framework designed to better capture multi-scale temporal semantics. The method decomposes time series into multiple temporal scales and introduces Multi-Scale Retrieval-Augmented Representations, where scale-aware prompts are constructed and used to retrieve semantically enriched representations from a large language model. The framework includes two main components: Scale-Aware Prompt Generation (SAPG), which uses dominant frequency information to construct scale-specific prompts, and Cross-Scale Semantic Disentanglement and Collaboration (CSDC), which aims to mitigate cross-scale interference while enabling collaboration among representations. A routing mechanism further selects the most relevant scale-specific representations for forecasting.

The paper targets an important challenge in LLM-augmented time series forecasting: the lack of explicit mechanisms for modeling multi-scale semantics and the potential interference between representations from different temporal granularities. The proposed design is coherent and empirically effective, demonstrating improvements across a broad set of datasets and forecasting settings. Ablation studies on the SAPG and CSDC modules further support the design choices, and experiments with larger LLM backbones suggest that the framework scales well with stronger language models.

**Compliance With Llm Reviewing Policy:**

Affirmed.

**Key Questions For Authors:**

- Is it possible to check the performance of replacing the LLM embeddings of p_n^s with purely numeric embeddings derived from x_n^s to compare the contribution of LLM and multi-scale information? I know the use of LLM is the key part and this can be an unreasonable experiment, but dividing the contribution of LLM and multi-scale information can help to understand the contribution of proposed method.

- What is the exact numeric tokenization in the prompts (rounding, normalization, delimiter choices)? Can you provide example prompts and discuss how token lengths scale with S and T?
- Can you report training/inference time and GPU memory for TimeMRA (GPT2-small and 7–8B backbones) relative to TimeCMA on representative datasets?

**Limitations:**

- Incomplete methodological details for reproducibility: Some components such as prompt construction, representation definitions, and routing training are not described with sufficient detail to allow reliable reproduction.
- Unclear dependence on LLM-derived representations: The paper does not clearly demonstrate whether the performance gains fundamentally depend on the LLM component or whether similar improvements could be achieved using purely numerical representations.

**Strengths And Weaknesses:**

# Strengths
- Well-motivated problem formuation

    The paper addresses an important limitation in existing LLM-augmented time series forecasting frameworks, namely the lack of explicit modeling for multi-scale semantics and the interference between representations across temporal scales. Tackling this issue is both practically relevant and timely given the increasing interest in combining LLMs with time series analysis.

- Intuitive model and prompt design

    Integration of global frequency analysis for period selection with scale-aware prompt generation is intuitive and well-motivated. The CSDC loss adopts a CLIP-style formulation to disentangle cross-scale representations as well as tie time series signals and LLM-derived signals, which aligns with the objective of utilizing the multi-scale information.

- Strong empirical results

    Experiment results across diverse datasets in various settings shows the performance improvement, and the ablation studies on the key modules verify their contributions.

# Weaknesses
- Limited clarity in notation and model description
    Several parts of the description are difficult to follow due to unclear notation and insufficient explanation. For example:

    -  The relationship between the representations E and $e_n$ is not clearly explained.
    -  The embedding process of $x_n^s$ is insufficiently describe.
    -  Details of prompt templates are minimal; concrete examples and token-length statistics per scale would help reproducibility and interpretability.

-  Unclear training mechanism for the Router Network

    The router network performs hard Top-M selection, but the paper does not clearly explain how this operation is trained in a differentiable manner. Without an explicit description of the training mechanism, it is difficult to understand how gradients propagate through the routing decision.

- Limited analysis of the LLM contribution

    While the method is framed as LLM-empowered forecasting, it remains unclear how critical the LLM component is to the final performance. A stronger ablation isolating the effect of LLM-derived representations would help clarify this. For instance, replacing the prompt-derived representations p_n^s with purely numeric embeddings derived from x_n^s could help evaluate whether the semantic information from the LLM is truly essential.

---

> ### Author Rebuttal · Authors · 2026-03-31
>
> We appreciate the reviewer's positive feedback, especially the recognition of our well-motivated problem formulation, intuitive design, and strong empirical results. For the constructive comments, we address each concern below.
>
> **W1&W2**: Explain notations and training mechanism.
>
> Thanks for your insightful comments and scientific rigor. $\hat{E} \in \mathbb{R}^{C \times N}$ represents the encoded variable features of all $N$ variables. $\hat{e}_n \in \hat{E}$ refers to the encoded features of the $n$th individual variable. $\tilde{\mathbf{X}}^s$ is derived by performing **multi-scale decomposition** and **linear mapping** on the input time series ${\mathbf{X}}$, and $\tilde{x}_n^s \in\tilde{\mathbf{X}}^s$ represents the temporal features of the $n$th variable at scale $s$.
>
> Following existing methods, we adopt the straight-through estimator to enable gradient propagation through the hard TopM selection. During the backward pass, the gradients are passed straight through the mask to the routing scores $g_n^s$​, which are differentiable with respect to the learnable parameters $W$.
>
> **W3&Q1:** Replacing LLM embeddings with numerical embeddings to assess the respective contributions of the LLM and multi-scale information.
>
> The ablation studies in Tables 4 and 12 of the original paper show that replacing the LLM with linear layers (-R LLM) performs worse than removing the multi-scale information (-w/o SAPG), **indicating that the use of the LLM is much more important than multi-scale information**. To further investigate the contribution of the LLM, we design the following variants:
>
> -w/o LLM: Removing the LLM and directly using the numeric embeddings as the augmented multi-scale representations.
>
> -RanInit: Replacing LLM with the randomly initialized Transformer of identical architecture and depth.
>
> Table 1
>
> |Methods|-w/o LLM|-RanInit|TimeMRA|
> |---|---|---|---|
> |Metric | MSE MAE |MSE MAE|MSE MAE|
> |96|0.411 0.407|0.371 0.379|**0.369 0.375**|
> |192|0.453 0.469|0.433 0.431|**0.428 0.423**|
> |336|0.500 0.473|0.437 0.440|**0.426 0.431**|
>
> The results on ETTh1 dataset (Table 1) show that: (1) -w/o LLM performs worst, the reason is that directly using numeric embeddings as augmented multi-scale representations **may introduce noise interference and lack specific multi-scale semantic information**. (2) TimeMRA outperforms -RanInit despite sharing the same architecture, confirming that the **pre-trained semantic knowledge is the genuine driver of performance**.
>
> **Q2**: What is the numeric tokenization in the prompts? Provide example prompts and discuss how token lengths scale with S and T?
>
> The input time series is pre-processed with RevIN, so **no additional normalization** is applied to the prompts. To ensure tokenization uniqueness and preserve clustering semantics, we retain only the integer part of the values. Commas and spaces are used as delimiters. A example of the prompt at scale $s$ for the $i$th variate is:
>
> "Those are the features of the $i$th variate at scale $s$, and the values are 1, -3, 0, 2, ..., -1. Extract the representative features."
>
> For a given variate at scale $s$, the subsequence length is $T^s = \lceil T / \wp^s \rceil$. In practice, as larger aggregation windows reduce $T^s$ at coarser scales, the total token length is dominated by the finest scale. A detailed token length statistics on ETTh1 dataset is given as follows:
>
> Table 2
>
> |Scale|Period $\wp^s$|$T^s$|Token Length|
> |---|---|---|---|
> |1|1|96|~220|
> |2|4|24|~76|
> |3|6|16|~60|
> |4|12|8|~44|
> |5|24|4|~36|
> |6|96|1|~30|
>
> As shown in Table 2, the total token count per variate is approximately 500, which is well within the 1024 token limit of GPT-2 Small.
>
> **Q3**: Report training/inference time and GPU memory of TimeMRA (GPT2-small and 7–8B backbones) relative to TimeCMA.
>
> We report the efficiency comparison on ETTh1 dataset with I=96 and O=96. All experiments are conducted on NVIDIA A100-80GB GPUs with a batch size of 32.
>
> Table 3
>
> |Methods|LLM Backbone|Training time (s)|Inference time (s)|GPU memory (MB)|MAE|
> |---|---|---|---|---|---|
> |TimeMRA|GPT2-small|73|**48**|**937**|0.375|
> |TimeCMA|GPT2-small|**49**|161|1,121| 0.391|
> |TimeMRA|LLaMA2-7B|3,105|1,908|21,354|**0.337**|
> |TimeCMA|LLaMA2-7B|1,910|6,334|26,705|0.370|
>
> The results in Table 3 show that: (1) LLM4TS methods **achieve better performance on more advanced LLM backbones**, but they require more training/inference time. (2) TimeMRA trains slower than TimeCMA, as TimeCMA **stores pre-computed last tokens** from the frozen LLM to avoid repetitive inference across epochs. (3) However, TimeMRA infers faster than TimeCMA. The reason is that inference requires only a single forward pass, the store strategy in TimeCMA offers limited benefits. In contrast, TimeMRA adopts lightweight non-LLM components (e.g., single-layer encoder/decoder), yielding lower inference latency. Overall, **considering the performance improvement and the computation cost, TimeMRA demonstrates superiority over TimeCMA**.

---

> > ### Author Rebuttal · Reviewer_C7pf · 2026-04-06
> >
> > Most of my concerns are resolved, and I decided to maintain my original score.
> > I recommend the authors to revise the description and explanation of the paper to reflect the answers to W1&2.

---

> > > ### Author Response · Authors · 2026-04-06
> > >
> > > Dear Reviewer C7pf,
> > >
> > > We are pleased that our responses have addressed your concerns! We **strongly agree with you that adding more detailed explanations and model descriptions will enhance the reproducibility and interpretability of our work**. All of your suggestions, including clarifications on notations, a more detailed explanation of the training mechanism for the router network, and ablation studies demonstrating the effectiveness of the LLM backbones, have been incorporated into the revised manuscript.
> > >
> > > Thanks again for the time and effort you dedicated to reviewing our paper. We would be deeply grateful for your further recognition and support.
> > >
> > > Best regards,
> > >
> > > The Authors

---

### Decision · Program_Chairs · 2026-04-30

**Decision:**

Accept (regular)

**Comment:**

The paper proposes TimeMRA, an LLM-empowered framework for time series forecasting designed to capture multi-scale temporal semantics. It specifically addresses two critical challenges in LLM-augmented time series forecasting: entangled multi-scale representations and redundant multi-scale interference.

Two reviewers provided positive scores, highlighting the paper's well-motivated problem formulation and well-organized presentation, and explicitly stated that the authors' rebuttal successfully resolved their concerns. The remaining two reviewers provided Weak Reject scores. However, neither submitted a final justification post-rebuttal. One reviewer failed to acknowledge the rebuttal and did not indicate whether the rebuttal has solved the concerns. This reviewer simply replied original score is kept without giving explicit reasons. The other reviewer acknoledged that the rebutall has partially resolved the concerns and have follow-up questions for the authors; but this reviewer did not respond, e.g., in the final justification, to indicate whether the additional follow-up questions are answered by the authors. After a careful and independent evaluation of the rebuttal and the authors' comprehensive responses, the AC feels that the authors have effectively addressed the raised concerns. Consequently, the overall recommendation from the AC tends to an Accept.